# Small nonlinearities in activation functions create bad local minima in neural networks

**Chulhee Yun, Suvrit Sra & Ali Jadbabaie**
Massachusetts Institute of Technology
Cambridge, MA 02139, USA
{`chulheey`,`suvrit`,`jadbabai`}@mit.edu

## Abstract

We investigate the loss surface of neural networks. We prove that even for one-hidden-layer networks with "slightest" nonlinearity, the empirical risks have spurious local minima in most cases. Our results thus indicate that in general "*no spurious local minima*" is a property limited to deep linear networks, and insights obtained from linear networks may not be robust. Specifically, for ReLU(-like) networks we constructively prove that for *almost all* practical datasets there exist infinitely many local minima. We also present a counterexample for more general activations (sigmoid, tanh, arctan, ReLU, etc.), for which there exists a bad local minimum. Our results make the least restrictive assumptions relative to existing results on spurious local optima in neural networks. We complete our discussion by presenting a comprehensive characterization of global optimality for deep linear networks, which unifies other results on this topic.

## 1 Introduction

Neural network training reduces to solving nonconvex empirical risk minimization problems, a task that is in general intractable. But success stories of deep learning suggest that local minima of the empirical risk could be close to global minima. Choromanska et al. (2015) use spherical spin-glass models from statistical physics to justify how the size of neural networks may result in local minima that are close to global. However, due to the complexities introduced by nonlinearity, a rigorous understanding of optimality in deep neural networks remains elusive.

Initial steps towards understanding optimality have focused on *deep linear* networks. This area has seen substantial recent progress. In deep linear networks there is no nonlinear activation; the output is simply a multilinear function of the input. Baldi & Hornik (1989) prove that some shallow networks have no spurious local minima, and Kawaguchi (2016) extends this result to squared error deep linear networks, showing that they only have global minima and saddle points. Several other works on linear nets have also appeared (Lu & Kawaguchi, 2017; Freeman & Bruna, 2017; Yun et al., 2018; Zhou & Liang, 2018; Laurent & Brecht, 2018a;b).

The theory of nonlinear neural networks (which is the actual setting of interest), however, is still in its infancy. There have been attempts to extend the "local minima are global" property from linear to nonlinear networks, but recent results suggest that this property does not usually hold (Zhou & Liang, 2018). Although not unexpected, rigorously proving such results turns out to be non-trivial, forcing several authors (e.g., Safran & Shamir (2018); Du et al. (2018b); Wu et al. (2018)) to make somewhat unrealistic assumptions (realizability and Gaussianity) on data.

In contrast, we prove existence of spurious local minima under the least restrictive (to our knowledge) assumptions. Since seemingly subtle changes to assumptions can greatly influence the analysis as well as the applicability of known results, let us first summarize what is known; this will also help provide a better intuitive perspective on our results (as the technical details are somewhat involved).

### 1.1 What is known so far?

There is a large and rapidly expanding literature of optimization of neural networks. Some works focus on the loss surface (Baldi & Hornik, 1989; Yu & Chen, 1995; Kawaguchi, 2016; Swirszcz et al., 2016; Soudry & Carmon, 2016; Xie et al., 2016; Nguyen & Hein, 2017; 2018; Safran & Shamir,

2018; Laurent & Brecht, 2018a; Yun et al., 2018; Zhou & Liang, 2018; Wu et al., 2018; Liang et al., 2018a;b; Shamir, 2018), while others study the convergence of gradient-based methods for optimizing this loss (Tian, 2017; Brutzkus & Globerson, 2017; Zhong et al., 2017; Soltanolkotabi, 2017; Li & Yuan, 2017; Du et al., 2018b; Zhang et al., 2018; Brutzkus et al., 2018; Wang et al., 2018; Li & Liang, 2018; Du et al., 2018a;c; Allen-Zhu et al., 2018; Zou et al., 2018; Zhou et al., 2019). In particular, our focus is on the loss surface itself, independent of any algorithmic concerns; this is reflected in the works summarized below.

For ReLU networks, the works (Swirszcz et al., 2016; Zhou & Liang, 2018) provide counterexample datasets that lead to spurious local minima, dashing hopes of "local implies global" properties. However, these works fail to provide statements about generic datasets, and one can argue that their setups are limited to isolated pathological examples. In comparison, our Theorem 1 shows existence of spurious local minima for *almost all* datasets, a much more general result. Zhou & Liang (2018) also give characterization of critical points of shallow ReLU networks, but with more than one hidden node the characterization provided is limited to certain regions.

There are also results that study population risk of shallow ReLU networks under an assumption that input data is i.i.d. Gaussian distributed (Safran & Shamir, 2018; Wu et al., 2018; Du et al., 2018b). Moreover, these works also assume *realizability*, i.e., the output data is generated from a neural network with the same architecture as the model one trains, with unknown true parameters. These assumptions enable one to compute the population risk in a closed form, and ensure that one can always achieve zero loss at global minima. The authors of Safran & Shamir (2018); Wu et al. (2018) study the population risk function of the form $\mathbb{E}_x[(\sum_{i=1}^{k} \text{ReLU}(w_i^T x) - \text{ReLU}(v_i^T x))^2]$, where the true parameters $v_i$'s are orthogonal unit vectors. Through extensive experiments and computer-assisted local minimality checks, Safran & Shamir (2018) show existence of local minima for $k \geq 6$. However, this result is empirical and does not have constructive proofs. Wu et al. (2018) show that with $k = 2$, there is no bad local minima on the manifold $\|w_1\|_2 = \|w_2\|_2 = 1$. Du et al. (2018b) study population risk of one-hidden-layer CNN. They show that there can be a spurious local minimum, but gradient descent converges to the global minimum with probability at least 1/4.

Our paper focuses on empirical risk instead of population risk, and *does not* assume either Gaussianity or realizability. Theorem 1 1's assumption on the dataset is that it is *not linearly fittable*[1], which is vastly more general and realistic than assuming that input data is Gaussian or that the output is generated from an unknown neural network. Our results also show that Wu et al. (2018) fails to extend to empirical risk and non-unit parameter vectors (see the discussion after Theorem 2).

Liang et al. (2018b) showed that under assumptions on the loss function, data distribution, network structure, and activation function, all local minima of the empirical loss have zero classification error in binary classification tasks. The result relies on stringent assumptions, and it is not directly comparable to ours because both "the local minimum has nonzero classification error" and "the local minima is spurious" do not imply one another. Liang et al. (2018a) proved that adding a parallel network with one exponential hidden node can eliminate all bad local minima. The result relies on the special parallel structure, whereas we analyze standard fully connected network architecture.

Laurent & Brecht (2018a) studies one-hidden-layer networks with hinge loss for classification. Under linear separability, the authors prove that Leaky-ReLU networks don't have bad local minima, while ReLU networks do. Our focus is on regression, and we only make mild assumptions on data.

For deep linear networks, the most relevant result to ours is Laurent & Brecht (2018b). When all hidden layers are wider than the input or output layers, Laurent & Brecht (2018b) prove that any local minimum of a deep linear network under differentiable convex loss is global.[2] They prove this by showing a statement about relationship between linear vs. multilinear parametrization. Our result in Theorem 4 is *strictly* more general that their results, and presents a comprehensive characterization.

A different body of literature (Yu & Chen, 1995; Soudry & Carmon, 2016; Xie et al., 2016; Nguyen & Hein, 2017; 2018) considers sufficient conditions for global optimality in nonlinear networks. These results make certain architectural assumptions (and some technical restrictions) that may not usually apply to realistic networks. There are also other works on global optimality conditions for specially designed architectures (Haeffele & Vidal, 2017; Feizi et al., 2017).

---

[1]That is, given input data matrices $X$ and $Y$, there is no matrix $R$ such that $Y = RX$.

[2]Although their result overlaps with a subset of Theorem 4, our theorem was obtained independently.

## 1.2 Contributions and Summary of Results

We summarize our key contributions more precisely below. Our work encompasses results for both nonlinear and linear neural networks. First, we study whether the "local minima are global" property holds for nonlinear networks. Unfortunately, our results here are negative. Specifically, we prove

▶ For piecewise linear and nonnegative homogeneous activation functions (e.g., ReLU), we prove in Theorem 1 that if linear models cannot perfectly fit the data, one can *construct* infinitely many local minima that are not global. In practice, most datasets are not linearly fittable, hence this result gives a constructive proof of spurious local minima for generic datasets. In contrast, several existing results either provide only one counterexample (Swirszcz et al., 2016; Zhou & Liang, 2018), or make restrictive assumptions of realizability (Safran & Shamir, 2018; Du et al., 2018b) or linear separability (Laurent & Brecht, 2018a). This result is presented in Section 2.

▶ In Theorem 2 we tackle more general nonlinear activation functions, and provide a simple architecture (with squared loss) and dataset, for which there exists a local minimum inferior to the global minimum for a realizable dataset. Our analysis applies to a wide range of activations, including sigmoid, tanh, arctan, ELU (Clevert et al., 2015), SELU (Klambauer et al., 2017), and ReLU. Considering that realizability of data simplifies the analysis and ensures zero loss at global optima, our counterexample that is realizable and yet has a spurious local minimum is surprising, suggesting that the situation is likely worse for non-realizable data. See Section 3 for details.

We complement our negative results by presenting the following positive result on linear networks:

▶ Assume that the hidden layers are as wide as either the input or the output, and that the empirical risk $\ell((W_j)_{j=1}^{H+1})$ equals $\ell_0(W_{H+1}W_H \cdots W_1)$, where $\ell_0$ is a differentiable loss function and $W_i$ is the weight matrix for layer $i$. Theorem 4 shows if $(\hat{W}_j)_{j=1}^{H+1}$ is a critical point of $\ell$, then its type of stationarity (local min/max, or saddle) is closely related to the behavior of $\ell_0$ evaluated at the product $\hat{W}_{H+1} \cdots \hat{W}_1$. If we additionally assume that any critical point of $\ell_0$ is a global minimum, Corollary 5 shows that the empirical risk $\ell$ only has global minima and saddles, and provides a simple condition to distinguish between them. To the best of our knowledge, this is the most general result on deep linear networks and it subsumes several previous results, e.g., (Kawaguchi, 2016; Yun et al., 2018; Zhou & Liang, 2018; Laurent & Brecht, 2018b). This result is in Section 4.

**Notation.** For an integer $a \geq 1$, $[a]$ denotes the set of integers from 1 to $a$ (inclusive). For a vector $v$, we use $[v]_i$ to denote its $i$-th component, while $[v]_{[i]}$ denotes a vector comprised of the first $i$ components of $v$. Let $\mathbf{1}_{(\cdot)}$ ($\mathbf{0}_{(\cdot)}$) be the all ones (zeros) column vector or matrix with size $(\cdot)$.

## 2 "ReLU-like" networks: bad local minima exist for most data

We study below whether nonlinear neural networks provably have spurious local minima. We show in §2 and §3 that even for extremely simple nonlinear networks, one encounters spurious local minima. We first consider ReLU and ReLU-like networks. Here, we prove that as long as linear models cannot perfectly fit the data, there exists a local minimum strictly inferior to the global one. Using nonnegative homogeneity, we can scale the parameters to get infinitely many local minima.

Consider a training dataset that consists of $m$ data points. The inputs and the outputs are of dimension $d_x$ and $d_y$, respectively. We aggregate these items, and write $X \in \mathbb{R}^{d_x \times m}$ as the data matrix and $Y \in \mathbb{R}^{d_y \times m}$ as the label matrix. Consider the 1-hidden-layer neural network $\hat{Y} = W_2 h(W_1 X + b_1 \mathbf{1}_m^T) + b_2 \mathbf{1}_m^T$, where $h$ is a nonlinear activation function, $W_2 \in \mathbb{R}^{d_y \times d_1}$, $b_2 \in \mathbb{R}^{d_y}$, $W_1 \in \mathbb{R}^{d_1 \times d_x}$, and $b_1 \in \mathbb{R}^{d_1}$. We analyze the empirical risk with squared loss

$$\ell(W_1, W_2, b_1, b_2) = \tfrac{1}{2}\|W_2 h(W_1 X + b_1 \mathbf{1}_m^T) + b_2 \mathbf{1}_m^T - Y\|_{\mathrm{F}}^2.$$

Next, define a class of piecewise linear nonnegative homogeneous functions

$$\bar{h}_{s_+, s_-}(x) = \max\{s_+ x, 0\} + \min\{s_- x, 0\}, \tag{1}$$

where $s_+ > 0, s_- \geq 0$ and $s_+ \neq s_-$. Note that ReLU and Leaky-ReLU are members of this class.

### 2.1 Main results and discussion

We use the shorthand $\tilde{X} := \begin{bmatrix} X^T & \mathbf{1}_m \end{bmatrix}^T \in \mathbb{R}^{(d_x+1) \times m}$. The main result of this section, Theorem 1, considers the case where linear models cannot fit $Y$, i.e., $Y \neq R\tilde{X}$ for all matrix $R$. With ReLU-like activation (1) and a few mild assumptions, Theorem 1 shows that there exist spurious local minima.

**Theorem 1.** *Suppose that the following conditions hold:*

*(C1.1) Output dimension is $d_y = 1$, and linear models $R\tilde{X}$ cannot perfectly fit $Y$.*

*(C1.2) All the data points $x_i$'s are distinct.*

*(C1.3) The activation function $h$ is $\bar{h}_{s_+,s_-}$.*

*(C1.4) The hidden layer has at least width 2: $d_1 \geq 2$.*

*Then, there is a spurious local minimum whose risk is the same as linear least squares model. Moreover, due to nonnegative homogeneity of $\bar{h}_{s_+,s_-}$, there are infinitely many such local minima.*

Noticing that most real world datasets cannot be perfectly fit with linear models, Theorem 1 shows that when we use the activation $\bar{h}_{s_+,s_-}$, the empirical risk has bad local minima for *almost all* datasets that one may encounter in practice. Although it is not very surprising that neural networks have spurious local minima, proving this rigorously is non-trivial. We provide a constructive and deterministic proof for this problem that holds for general datasets, which is in contrast to experimental results of Safran & Shamir (2018). We emphasize that Theorem 1 also holds even for "slightest" nonlinearities, e.g., when $s_+ = 1 + \epsilon$ and $s_- = 1$ where $\epsilon > 0$ is small. This suggests that the "local min is global" property is limited to the simplified setting of *linear* neural networks.

Existing results on squared error loss either provide one counterexample (Swirszcz et al., 2016; Zhou & Liang, 2018), or assume realizability and Gaussian input (Safran & Shamir, 2018; Du et al., 2018b). Realizability is an assumption that the output is generated by a network with unknown parameters. In real datasets, neither input is Gaussian nor output is generated by neural networks; in contrast, our result holds for most realistic situations, and hence delivers useful insight.

There are several results proving sufficient conditions for global optimality of nonlinear neural networks (Soudry & Carmon, 2016; Xie et al., 2016; Nguyen & Hein, 2017). But they rely on assumptions that the network width scales with the number of data points. For instance, applying Theorem 3.4 of Nguyen & Hein (2017) to our network proves that if $\tilde{X}$ has linearly independent columns and other assumptions hold, then any critical point with $W_2 \neq 0$ is a global minimum. However, linearly independent columns already imply $\text{row}(\tilde{X}) = \mathbb{R}^m$, so even linear models $R\tilde{X}$ can fit any $Y$; i.e., there is less merit in using a complex model to fit $Y$. Theorem 1 does not make any structural assumption other than $d_1 \geq 2$, and addresses the case where it is *impossible* to fit $Y$ with linear models, which is much more realistic.

It is worth comparing our result with Laurent & Brecht (2018a), who use hinge loss based classification and assume linear separability to prove "no spurious local minima" for Leaky-ReLU networks. Their result does not contradict our theorem because the losses are different and we do not assume linear separability.

One might wonder if our theorem holds even with $d_1 \geq m$. Venturi et al. (2018) showed that one-hidden-layer neural networks with $d_1 \geq m$ doesn't have spurious valleys, hence there is no *strict* spurious local minima; however, due to nonnegative homogeneity of $\bar{h}_{s_+,s_-}$ we only have non-strict local minima. Based on Bengio et al. (2006), one might claim that with wide enough hidden layer and random $W_1$ and $b_1$, one can fit any $Y$; however, this is not the case, by our assumption that linear models $R\tilde{X}$ cannot fit $Y$. Note that for any $d_1$, there is a non-trivial region (measure $> 0$) in the parameter space where $W_1 X + b_1 \mathbf{1}_m^T > \mathbf{0}$ (entry-wise). In this region, the output of neural network $\hat{Y}$ is still a linear combination of rows of $\tilde{X}$, so $\hat{Y}$ cannot fit $Y$; in fact, it can only do as well as linear models. We will see in the Step 1 of Section 2.2 that the bad local minimum that we construct "kills" $d_1 - 1$ neurons; however, killing many neurons is not a necessity, and it is just to simply the exposition. In fact, any local minimum in the region $W_1 X + b_1 \mathbf{1}_m^T > \mathbf{0}$ is a spurious local minimum.

## 2.2 ANALYSIS OF THEOREM 1

The proof of the theorem is split into two steps. First, we prove that there exist local minima $(\hat{W}_j, \hat{b}_j)_{j=1}^2$ whose risk value is the same as the linear least squares solution, and that there are infinitely many such minima. Second, we will construct a tuple of parameters $(\tilde{W}_j, \tilde{b}_j)_{j=1}^2$ that has strictly smaller empirical risk than $(\hat{W}_j, \hat{b}_j)_{j=1}^2$.

**Step 1: A local minimum as good as the linear solution.** The main idea here is to exploit the weights from the linear least squares solution, and to tune the parameters so that all inputs to hidden nodes become positive. Doing so makes the hidden nodes "locally linear," so that the constructed $(\hat{W}_j, \hat{b}_j)_{j=1}^2$ that produce linear least squares estimates at the output become locally optimal.

Recall that $\tilde{X} = \begin{bmatrix} X^T & \mathbf{1}_m \end{bmatrix}^T \in \mathbb{R}^{(d_x+1)\times m}$, and define a linear least squares loss $\ell_0(R) := \frac{1}{2}\|R\tilde{X} - Y\|_F^2$ that is minimized at $\bar{W}$, so that $\nabla\ell_0(\bar{W}) = (\bar{W}\tilde{X} - Y)\tilde{X}^T = 0$. Since $d_y = 1$, the solution $\bar{W} \in \mathbb{R}^{d_y \times (d_x+1)}$ is a row vector. For all $i \in [m]$, let $\bar{y}_i = \bar{W}\begin{bmatrix} x_i^T & 1 \end{bmatrix}^T$ be the output of the linear least squares model, and similarly $\bar{Y} = \bar{W}\tilde{X}$.

Let $\eta := \min\{-1, 2\min_i \bar{y}_i\}$, a negative constant making $\bar{y}_i - \eta > 0$ for all $i$. Define parameters

$$\hat{W}_1 = \alpha\begin{bmatrix} [\bar{W}]_{[d_x]} \\ \mathbf{0}_{(d_1-1)\times d_x} \end{bmatrix}, \; \hat{b}_1 = \alpha\begin{bmatrix} [\bar{W}]_{d_x+1} - \eta \\ -\eta\mathbf{1}_{d_1-1} \end{bmatrix}, \; \hat{W}_2 = \begin{bmatrix} \frac{1}{\alpha s_+} & \mathbf{0}_{d_1-1}^T \end{bmatrix}, \; \hat{b}_2 = \eta,$$

where $\alpha > 0$ is any arbitrary fixed positive constant, $[\bar{W}]_{[d_x]}$ gives the first $d_x$ components of $\bar{W}$, and $[\bar{W}]_{d_x+1}$ the last component. Since $\bar{y}_i = [\bar{W}]_{[d_x]}x_i + [\bar{W}]_{d_x+1}$, for any $i$, $\hat{W}_1 x_i + \hat{b}_1 > \mathbf{0}_{d_1}$ (component-wise), given our choice of $\eta$. Thus, all hidden node inputs are positive. Moreover, $\hat{Y} = \frac{1}{\alpha s_+}s_+(\alpha\bar{Y} - \alpha\eta\mathbf{1}_m^T) + \eta\mathbf{1}_m^T = \bar{Y}$, so that the loss $\ell((\hat{W}_j, \hat{b}_j)_{j=1}^2) = \frac{1}{2}\|\bar{Y} - Y\|_F^2 = \ell_0(\bar{W})$.

So far, we checked that $(\hat{W}_j, \hat{b}_j)_{j=1}^2$ has the same empirical risk as a linear least squares solution. It now remains to show that this point is indeed a local minimum of $\ell$. To that end, we consider the perturbed parameters $(\hat{W}_j + \Delta_j, \hat{b}_j + \delta_j)_{j=1}^2$, and check their risk is always larger. A useful point is that since $\bar{W}$ is a minimum of $\ell_0(R) = \frac{1}{2}\|R\tilde{X} - Y\|_F^2$, we have

$$(\bar{W}\tilde{X} - Y)\tilde{X}^T = (\bar{Y} - Y)\begin{bmatrix} X^T & \mathbf{1}_m \end{bmatrix} = 0, \tag{2}$$

so $(\bar{Y} - Y)X^T = 0$ and $(\bar{Y} - Y)\mathbf{1}_m = 0$. For small enough perturbations, $(\hat{W}_1 + \Delta_1)x_i + (\hat{b}_1 + \delta_1) > 0$ still holds for all $i$. So, we can observe that

$$\ell((\hat{W}_j + \Delta_j, \hat{b}_j + \delta_j)_{j=1}^2) = \frac{1}{2}\|\bar{Y} - Y + \tilde{\Delta}X + \tilde{\delta}\mathbf{1}_m^T\|_F^2 = \frac{1}{2}\|\bar{Y} - Y\|_F^2 + \frac{1}{2}\|\tilde{\Delta}X + \tilde{\delta}\mathbf{1}_m^T\|_F^2, \tag{3}$$

where $\tilde{\Delta}$ and $\tilde{\delta}$ are $\tilde{\Delta} := s_+(\hat{W}_2\Delta_1 + \Delta_2\hat{W}_1 + \Delta_2\Delta_1)$ and $\tilde{\delta} := s_+(\hat{W}_2\delta_1 + \Delta_2\hat{b}_1 + \Delta_2\delta_1) + \delta_2$; they are aggregated perturbation terms. We used (2) to obtain the last equality of (3). Thus, $\ell((\hat{W}_j + \Delta_j, \hat{b}_j + \delta_j)_{j=1}^2) \geq \ell((\hat{W}_j, \hat{b}_j)_{j=1}^2)$ for small perturbations, proving $(\hat{W}_j, \hat{b}_j)_{j=1}^2$ is indeed a local minimum of $\ell$. Since this is true for arbitrary $\alpha > 0$, there are infinitely many such local minima. We can also construct similar local minima by permuting hidden nodes, etc.

**Step 2: A point strictly better than the local minimum.** The proof of this step is more involved. In the previous step, we "pushed" all the input to the hidden nodes to positive side, and took advantage of "local linearity" of the hidden nodes near $(\hat{W}_j, \hat{b}_j)_{j=1}^2$. But to construct parameters $(\tilde{W}_j, \tilde{b}_j)_{j=1}^2$ that have strictly smaller risk than $(\hat{W}_j, \hat{b}_j)_{j=1}^2$ (to prove that $(\hat{W}_j, \hat{b}_j)_{j=1}^2$ is a spurious local minimum), we make the sign of inputs to the hidden nodes different depending on data.

To this end, we sort the indices of data points in increasing order of $\bar{y}_i$; i.e., $\bar{y}_1 \leq \bar{y}_2 \leq \cdots \leq \bar{y}_m$. Define the set $\mathcal{J} := \{j \in [m-1] \mid \sum_{i\leq j}(\bar{y}_i - y_i) \neq 0, \bar{y}_j < \bar{y}_{j+1}\}$. The remaining construction is divided into two cases: $\mathcal{J} \neq \emptyset$ and $\mathcal{J} = \emptyset$, whose main ideas are essentially the same. We present the proof for $\mathcal{J} \neq \emptyset$, and defer the other case to Appendix A2 as it is rarer, and its proof, while instructive for its perturbation argument, is technically too involved.

**Case 1: $\mathcal{J} \neq \emptyset$.** Pick any $j_0 \in \mathcal{J}$. We can observe that $\sum_{i\leq j_0}(\bar{y}_i - y_i) = -\sum_{i>j_0}(\bar{y}_i - y_i)$, because of (2). Define $\beta = \frac{\bar{y}_{j_0} + \bar{y}_{j_0+1}}{2}$, so that $\bar{y}_i - \beta < 0$ for all $i \leq j_0$ and $\bar{y}_i - \beta > 0$ for all $i > j_0$. Then, let $\gamma$ be a constant satisfying $0 < |\gamma| \leq \frac{\bar{y}_{j_0+1} - \bar{y}_{j_0}}{4}$, whose value will be specified later. Since $|\gamma|$ is small enough, $\text{sign}(\bar{y}_i - \beta) = \text{sign}(\bar{y}_i - \beta + \gamma) = \text{sign}(\bar{y}_i - \beta - \gamma)$. Now select parameters

$$\tilde{W}_1 = \begin{bmatrix} [\bar{W}]_{[d_x]} \\ -[\bar{W}]_{[d_x]} \\ \mathbf{0}_{(d_1-2)\times d_x} \end{bmatrix}, \; \tilde{b}_1 = \begin{bmatrix} [\bar{W}]_{d_x+1} - \beta + \gamma \\ -[\bar{W}]_{d_x+1} + \beta + \gamma \\ \mathbf{0}_{d_1-2} \end{bmatrix}, \; \tilde{W}_2 = \frac{1}{s_+ + s_-}\begin{bmatrix} 1 & -1 & \mathbf{0}_{d_1-2}^T \end{bmatrix}, \; \tilde{b}_2 = \beta.$$

Recall again that $[\bar{W}]_{[d_x]}x_i + [\bar{W}]_{d_x+1} = \bar{y}_i$. For $i \leq j_0$, $\bar{y}_i - \beta + \gamma < 0$ and $-\bar{y}_i + \beta + \gamma > 0$, so

$$\hat{y}_i = \frac{s_-(\bar{y}_i - \beta + \gamma)}{s_+ + s_-} - \frac{s_+(-\bar{y}_i + \beta + \gamma)}{s_+ + s_-} + \beta = \bar{y}_i - \frac{s_+ - s_-}{s_+ + s_-}\gamma.$$

Similarly, for $i > j_0$, $\bar{y}_i - \beta + \gamma > 0$ and $-\bar{y}_i + \beta + \gamma < 0$ results in $\hat{y}_i = \bar{y}_i + \frac{s_+ - s_-}{s_+ + s_-}\gamma$. Here, we push the outputs $\hat{y}_i$ of the network by $\frac{s_+ - s_-}{s_+ + s_-}\gamma$ from $\bar{y}_i$, and the direction of the "push" varies depending on whether $i \leq j_0$ or $i > j_0$.

The empirical risk for this choice of parameters is

$$\ell((\tilde{W}_j, \tilde{b}_j)_{j=1}^2) = \frac{1}{2}\sum_{i \leq j_0}\left(\bar{y}_i - \frac{s_+ - s_-}{s_+ + s_-}\gamma - y_i\right)^2 + \frac{1}{2}\sum_{i > j_0}\left(\bar{y}_i + \frac{s_+ - s_-}{s_+ + s_-}\gamma - y_i\right)^2$$

$$= \ell_0(\bar{W}) - 2\left[\sum_{i \leq j_0}(\bar{y}_i - y_i)\right]\frac{s_+ - s_-}{s_+ + s_-}\gamma + O(\gamma^2).$$

Since $\sum_{i \leq j_0}(\bar{y}_i - y_i) \neq 0$ and $s_+ \neq s_-$, we can choose $\text{sign}(\gamma) = \text{sign}([\sum_{i \leq j_0}(\bar{y}_i - y_i)](s_+ - s_-))$, and choose small $|\gamma|$ so that $\ell((\tilde{W}_j, \tilde{b}_j)_{j=1}^2) < \ell_0(\bar{W}) = \ell((\hat{W}_j, \hat{b}_j)_{j=1}^2)$, proving that $(\hat{W}_j, \hat{b}_j)_{j=1}^2$ is a spurious local minimum.

# 3 COUNTEREXAMPLE: BAD LOCAL MINIMA FOR MANY ACTIVATIONS

The proof of Theorem 1 crucially exploits the piecewise linearity of the activation functions. Thus, one may wonder whether the spurious local minima seen there are an artifact of the specific nonlinearity. We show below that this is *not* the case. We provide a counterexample nonlinear network and a dataset for which a wide range of nonlinear activations result in a local minimum that is strictly inferior to the global minimum with exactly zero empirical risk. Examples of such activation functions include popular activation functions such as sigmoid, tanh, arctan, ELU, SELU, and ReLU.

We consider again the squared error empirical risk of a one-hidden-layer nonlinear neural network:

$$\ell((W_j, b_j)_{j=1}^2) := \frac{1}{2}\|W_2 h(W_1 X + b_1 \mathbf{1}_m^T) + b_2 \mathbf{1}_m^T - Y\|_F^2,$$

where we fix $d_x = d_1 = 2$ and $d_y = 1$. Also, let $h^{(k)}(x)$ be the $k$-th derivative of $h : \mathbb{R} \mapsto \mathbb{R}$, whenever it exists at $x$. For short, let $h'$ and $h''$ denote the first and second derivatives.

## 3.1 MAIN RESULTS AND DISCUSSION

**Theorem 2.** *Let the loss $\ell((W_j, b_j)_{j=1}^2)$ and network be as defined above. Consider the dataset*

$$X = \begin{bmatrix} 1 & 0 & \frac{1}{2} \\ 0 & 1 & \frac{1}{2} \end{bmatrix}, Y = \begin{bmatrix} 0 & 0 & 1 \end{bmatrix}.$$

*For this network and dataset the following results hold:*

*1. If there exist real numbers $v_1, v_2, v_3, v_4 \in \mathbb{R}$ such that*

    *(C2.1) $h(v_1)h(v_4) = h(v_2)h(v_3)$, and*
    *(C2.2) $h(v_1)h\left(\frac{v_3+v_4}{2}\right) \neq h(v_3)h\left(\frac{v_1+v_2}{2}\right)$,*

    *then there is a tuple $(\tilde{W}_j, \tilde{b}_j)_{j=1}^2$ at which $\ell$ equals $0$.*

*2. If there exist real numbers $v_1, v_2, u_1, u_2 \in \mathbb{R}$ such that the following conditions hold:*

    *(C2.3) $u_1 h(v_1) + u_2 h(v_2) = \frac{1}{3}$,*
    *(C2.4) $h$ is infinitely differentiable at $v_1$ and $v_2$.*
    *(C2.5) there exists a constant $c > 0$ such that $|h^{(n)}(v_1)| \leq c^n n!$ and $|h^{(n)}(v_2)| \leq c^n n!$.*
    *(C2.6) $(u_1 h'(v_1))^2 + \frac{u_1 h''(v_1)}{3} > 0$,*
    *(C2.7) $(u_1 h'(v_1) u_2 h'(v_2))^2 < ((u_1 h'(v_1))^2 + \frac{u_1 h''(v_1)}{3})((u_2 h'(v_2))^2 + \frac{u_2 h''(v_2)}{3})$,*

    *then there exists a tuple $(\hat{W}_j, \hat{b}_j)_{j=1}^2$ such that the output of the network is the same as the linear least squares model, the risk $\ell((\hat{W}_j, \hat{b}_j)_{j=1}^2) = \frac{1}{3}$, and $(\hat{W}_j, \hat{b}_j)_{j=1}^2$ is a local minimum of $\ell$.*

Theorem 2 shows that for this architecture and dataset, activations that satisfy (C2.1)–(C2.7) introduce at least one spurious local minimum. Notice that the empirical risk is zero at the global minimum. This means that the data $X$ and $Y$ can actually be "generated" by the network, which satisfies the realizability assumption that others use (Safran & Shamir, 2018; Du et al., 2018b; Wu et al., 2018). Notice that our counterexample is "easy to fit," and yet, there exists a local minimum that is not global. This leads us to conjecture that with harder datasets, the problems with spurious local minima could be worse. The proof of Theorem 2 can be found in Appendix A3.

**Discussion.** Note that the conditions (C2.1)–(C2.7) only require *existence* of certain real numbers rather than some *global* properties of activation $h$, hence are not as restrictive as they look. Conditions (C2.1)–(C2.2) come from a choice of tuple $(\tilde{W}_j, \tilde{b}_j)_{j=1}^2$ that perfectly fits the data. Condition (C2.3) is necessary for constructing $(\hat{W}_j, \hat{b}_j)_{j=1}^2$ with the same output as the linear least squares model, and Conditions (C2.4)–(C2.7) are needed for showing local minimality of $(\hat{W}_j, \hat{b}_j)_{j=1}^2$ via Taylor expansions. The class of functions that satisfy conditions (C2.1)–(C2.7) is quite large, and includes the nonlinear activation functions used in practice. The next corollary highlights this observation (for a proof with explicit choices of the involved real numbers, please see Appendix A5).

**Corollary 3.** *For the counterexample in Theorem 2, the set of activation functions satisfying conditions (C2.1)–(C2.7) include sigmoid, tanh, arctan, quadratic, ELU, and SELU.*

Admittedly, Theorem 2 and Corollary 3 give one counterexample instead of stating a claim about generic datasets. Nevertheless, this example shows that for many practical nonlinear activations, the desirable "local minimum is global" property cannot hold even for realizable datasets, suggesting that the situation could be worse for non-realizable ones.

**Remark: "ReLU-like" activation functions.** Recall the piecewise linear nonnegative homogeneous activation function $\bar{h}_{s_+,s_-}$. They do not satisfy condition (C2.7), so Theorem 2 cannot be directly applied. Also, if $s_- = 0$ (i.e., ReLU), conditions (C2.1)–(C2.2) are also violated. However, the statements of Theorem 2 hold even for $\bar{h}_{s_+,s_-}$, which is shown in Appendix A6. Recalling again $s_+ = 1 + \epsilon$ and $s_- = 1$, this means that even with the "slightest" nonlinearity in activation function, the network has a global minimum with risk zero while there exists a bad local minimum that performs just as linear least squares models. In other words, "local minima are global" property is rather brittle and can only hold for linear neural networks. Another thing to note is that in Appendix A6, the bias parameters are all zero, for both $(\tilde{W}_j, \tilde{b}_j)_{j=1}^2$ and $(\hat{W}_j, \hat{b}_j)_{j=1}^2$. For models without bias parameters, $(\hat{W}_j)_{j=1}^2$ is still a spurious local minimum, thus showing that Wu et al. (2018) fails to extend to empirical risks and non-unit weight vectors.

## 4  GLOBAL OPTIMALITY IN LINEAR NETWORKS

In this section we present our results on deep linear neural networks. Assuming that the hidden layers are at least as wide as either the input or output, we show that critical points of the loss with a multilinear parameterization inherit the type of critical points of the loss with a linear parameterization. As a corollary, we show that for differentiable losses whose critical points are globally optimal, deep linear networks have *only global minima or saddle points*. Furthermore, we provide an efficiently checkable condition for global minimality.

Suppose the network has $H$ hidden layers having widths $d_1, \ldots, d_H$. To ease notation, we set $d_0 = d_x$ and $d_{H+1} = d_y$. The weights between adjacent layers are kept in matrices $W_j \in \mathbb{R}^{d_j \times d_{j-1}}$ ($j \in [H+1]$), and the output $\hat{Y}$ of the network is given by the product of weight matrices with the data matrix: $\hat{Y} = W_{H+1} W_H \cdots W_1 X$. Let $(W_j)_{j=1}^{H+1}$ be the tuple of all weight matrices, and $W_{i:j}$ denote the product $W_i W_{i-1} \cdots W_{j+1} W_j$ for $i \geq j$, and the identity for $i = j - 1$. We consider the empirical risk $\ell((W_j)_{j=1}^{H+1})$, which, for linear networks assumes the form

$$\ell((W_j)_{j=1}^{H+1}) := \ell_0(W_{H+1:1}), \tag{4}$$

where $\ell_0$ is a suitable differentiable loss. For example, when $\ell_0(R) = \frac{1}{2}\|RX - Y\|_{\mathrm{F}}^2$, $\ell((W_j)_{j=1}^{H+1}) = \frac{1}{2}\|W_{H+1:1}X - Y\|_{\mathrm{F}}^2 = \ell_0(W_{H+1:1})$. Lastly, we write $\nabla \ell_0(M) \equiv \nabla_R \ell_0(R)|_{R=M}$.

**Remark: bias terms.** We omit the bias terms $b_1, \ldots, b_{H+1}$ here. This choice is for simplicity; models with bias can be handled by the usual trick of augmenting data and weight matrices.

## 4.1 Main results and discussion

We are now ready to state our first main theorem, whose proof is deferred to Appendix A7.

**Theorem 4.** *Suppose that for all $j$, $d_j \geq \min\{d_x, d_y\}$, and that the loss $\ell$ is given by (4), where $\ell_0$ is differentiable on $\mathbb{R}^{d_y \times d_x}$. For any critical point $(\hat{W}_j)_{j=1}^{H+1}$ of the loss $\ell$, the following claims hold:*

1. *If $\nabla \ell_0(\hat{W}_{H+1:1}) \neq 0$, then $(\hat{W}_j)_{j=1}^{H+1}$ is a saddle of $\ell$.*

2. *If $\nabla \ell_0(\hat{W}_{H+1:1}) = 0$, then*

    (a) *$(\hat{W}_j)_{j=1}^{H+1}$ is a local min (max) of $\ell$ if $\hat{W}_{H+1:1}$ is a local min (max) of $\ell_0$; moreover,*

    (b) *$(\hat{W}_j)_{j=1}^{H+1}$ is a global min (max) of $\ell$ if and only if $\hat{W}_{H+1:1}$ is a global min (max) of $\ell_0$.*

3. *If there exists $j^* \in [H+1]$ such that $\hat{W}_{H+1:j^*+1}$ has full row rank and $\hat{W}_{j^*-1:1}$ has full column rank, then $\nabla \ell_0(\hat{W}_{H+1:1}) = 0$, so 2(a) and 2(b) hold. Also,*

    (a) *$\hat{W}_{H+1:1}$ is a local min (max) of $\ell_0$ if $(\hat{W}_j)_{j=1}^{H+1}$ is a local min (max) of $\ell$.*

Let us paraphrase Theorem 4 in words. In particular, it states that if the hidden layers are "wide enough" so that the product $W_{H+1:1}$ can attain full rank and if the loss $\ell$ assumes the form (4) for a differentiable loss $\ell_0$, then the type (optimal or saddle point) of a critical point $(\hat{W}_j)_{j=1}^{H+1}$ of $\ell$ is governed by the behavior of $\ell_0$ at the product $\hat{W}_{H+1:1}$.

Note that for any critical point $(\hat{W}_j)_{j=1}^{H+1}$ of the loss $\ell$, either $\nabla \ell_0(\hat{W}_{H+1:1}) \neq 0$ or $\nabla \ell_0(\hat{W}_{H+1:1}) = 0$. Parts 1 and 2 handle these two cases. Also observe that the condition in Part 3 implies $\nabla \ell_0 = 0$, so Part 3 is a refinement of Part 2. A notable fact is that a sufficient condition for Part 3 is $\hat{W}_{H+1:1}$ having full rank. For example, if $d_x \geq d_y$, full-rank $\hat{W}_{H+1:1}$ implies $\operatorname{rank}(\hat{W}_{H+1:2}) = d_y$, whereby the condition in Part 3 holds with $j^* = 1$.

If $\hat{W}_{H+1:1}$ is not critical for $\ell_0$, then $(\hat{W}_j)_{j=1}^{H+1}$ must be a saddle point of $\ell$. If $\hat{W}_{H+1:1}$ is a local min/max of $\ell_0$, $(\hat{W}_j)_{j=1}^{H+1}$ is also a local min/max of $\ell$. Notice, however, that Part 2(a) does not address the case of saddle points; when $\hat{W}_{H+1:1}$ is a saddle point of $\ell_0$, the tuple $(\hat{W}_j)_{j=1}^{H+1}$ can behave arbitrarily. However, with the condition in Part 3, statements 2(a) and 3(a) hold at the same time, so that $\hat{W}_{H+1:1}$ is a local min/max of $\ell_0$ *if and only if* $(\hat{W}_j)_{j=1}^{H+1}$ is a local min/max of $\ell$. Observe that the same "if and only if" statement holds for saddle points due to their definition; in summary, the types (min/max/saddle) of the critical points $(\hat{W}_j)_{j=1}^{H+1}$ and $\hat{W}_{H+1:1}$ match exactly.

Although Theorem 4 itself is of interest, the following corollary highlights its key implication for deep linear networks.

**Corollary 5.** *In addition to the assumptions in Theorem 4, assume that any critical point of $\ell_0$ is a global min (max). For any critical point $(\hat{W}_j)_{j=1}^{H+1}$ of $\ell$, if $\nabla \ell_0(\hat{W}_{H+1:1}) \neq 0$, then $(\hat{W}_j)_{j=1}^{H+1}$ is a saddle of $\ell$, while if $\nabla \ell_0(\hat{W}_{H+1:1}) = 0$, then $(\hat{W}_j)_{j=1}^{H+1}$ is a global min (max) of $\ell$.*

**Proof**   If $\nabla \ell_0(\hat{W}_{H+1:1}) \neq 0$, then $\hat{W}_{H+1:1}$ is a saddle point by Theorem 4.1. If $\nabla \ell_0(\hat{W}_{H+1:1}) = 0$, then $\hat{W}_{H+1:1}$ is a global min (max) of $\ell_0$ by assumption. By Theorem 4.2(b), $(\hat{W}_j)_{j=1}^{H+1}$ must be a global min (max) of $\ell$. ☐

Corollary 5 shows that for any differentiable loss function $\ell_0$ whose critical points are global minima, the loss $\ell$ has only global minima and saddle points, therefore satisfying the "local minima are global" property. In other words, for such an $\ell_0$, the multilinear re-parametrization introduced by deep linear networks *does not introduce any spurious local minima/maxima*; it only introduces saddle points. Importantly, Corollary 5 also provides a checkable condition that distinguishes global minima from saddle points. Since $\ell$ is nonconvex, it is remarkable that such a simple necessary and sufficient condition for global optimality is available.

Our result generalizes previous works on linear networks such as Kawaguchi (2016); Yun et al. (2018); Zhou & Liang (2018), because it provides conditions for global optimality for a broader range of loss functions without assumptions on datasets. Laurent & Brecht (2018b) proved that if

$(\hat{W}_j)_{j=1}^{H+1}$ is a local min of $\ell$, then $\hat{W}_{H+1:1}$ is a critical point of $\ell_0$. First, observe that this result is implied by Theorem 4.1. So our result, which was proved in parallel and independently, is strictly more general. With additional assumption that critical points of $\ell_0$ are global minima, Laurent & Brecht (2018b) showed that "local min is global" property holds for linear neural networks; our Corollay 5 gives a simple and efficient test condition as well as proving there are only global minima and saddles, which is clearly stronger.

## 5 DISCUSSION AND FUTURE WORK

We investigated the loss surface of deep linear and nonlinear neural networks. We proved two theorems showing existence of spurious local minima on nonlinear networks, which apply to almost all datasets (Theorem 1) and a wide class of activations (Theorem 2). We concluded by Theorem 4, showing a general result studying the behavior of critical points in multilinearly parametrized functions, which unifies other existing results on linear neural networks. Given that spurious local minima are common in neural networks, a valuable future research direction will be investigating how far local minima are from global minima in general, and how the size of the network affects this gap. Another thing to note is that even though we showed the existence of spurious local minima in the *whole* parameter space, things can be different in restricted sets of parameter space (e.g., by adding regularizers). Understanding the loss surface in such sets would be valuable. Additionally, one can try to show algorithmic/trajectory results of (stochastic) gradient descent. We hope that our paper will be a stepping stone to such future research.

ACKNOWLEDGMENTS

This work was supported by the DARPA Lagrange Program. Suvrit Sra also acknowledges support from an Amazon Research Award.

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

## A1 NOTATION

We first list notation used throughout the appendix. For integers $a \leq b$, $[a, b]$ denotes the set of integers between them. We write $[b]$, if $a = 1$. For a vector $v$, we use $[v]_i$ to denote its $i$-th component, while $[v]_{[i]}$ denotes a vector comprised of the first $i$ components of $v$. Let $\mathbf{1}_d$ (or $\mathbf{0}_d$) be the all ones (zeros) column vector in $\mathbb{R}^d$. For a subspace $V \subseteq \mathbb{R}^d$, we denote by $V^{\perp}$ its orthogonal complement.

For a matrix $A$, $[A]_{i,j}$ is the $(i, j)$-th entry and $[A]_{\cdot,j}$ its $j$-th column. Let $\sigma_{\max}(A)$ and $\sigma_{\min}(A)$ denote the largest and smallest singular values of $A$, respectively; $\mathrm{row}(A)$, $\mathrm{col}(A)$, $\mathrm{rank}(A)$, and $\|A\|_{\mathrm{F}}$ denote respectively the row space, column space, rank, and Frobenius norm of matrix $A$. Let $\mathrm{null}(A) := \{v \mid Av = 0\}$ and $\mathrm{leftnull}(A) := \{v \mid v^T A = 0\}$ be the null space and the left-null space of $A$, respectively. When $A$ is a square matrix, let $\mathrm{tr}(A)$ be the trace of $A$. For matrices $A$ and $B$ of the same size, $\langle A, B \rangle = \mathrm{tr}(A^T B)$ denotes the usual trace inner product of $A$ and $B$. Equivalently, $\langle A, B \rangle = \mathrm{tr}(A^T B) = \mathrm{tr}(AB^T)$. Let $\mathbf{0}_{d \times m}$ be the all zeros matrix in $\mathbb{R}^{d \times m}$.

## A2 PROOF OF THEOREM 1, STEP 2, CASE 2

**Case 2.** $\mathcal{J} = \emptyset$. We start with a lemma discussing what $\mathcal{J} = \emptyset$ implies.

**Lemma A.1.** *If $\mathcal{J} = \emptyset$, the following statements hold:*

1. *There are some $\bar{y}_j$'s that are duplicate; i.e. for some $i \neq j$, $\bar{y}_i = \bar{y}_j$.*

2. *If $\bar{y}_j$ is non-duplicate, meaning that $\bar{y}_{j-1} < \bar{y}_j < \bar{y}_{j+1}$, $\bar{y}_j = y_j$ holds.*

3. *If $\bar{y}_j$ is duplicate, $\sum_{i:\bar{y}_i = \bar{y}_j}(\bar{y}_i - y_i) = 0$ holds.*

4. *There exists at least one duplicate $\bar{y}_j$ such that, for that $\bar{y}_j$, there exist at least two different $i$'s that satisfy $\bar{y}_i = \bar{y}_j$ and $\bar{y}_i \neq y_i$.*

**Proof** We prove this by showing if any of these statements are not true, then we have $\mathcal{J} \neq \emptyset$ or a contradiction.

1. If all the $\bar{y}_j$'s are distinct and $\mathcal{J} = \emptyset$, by definition of $\mathcal{J}$, $\bar{y}_j = y_j$ for all $j$. This violates our assumption that linear models cannot perfectly fit $Y$.

2. If we have $\bar{y}_j \neq y_j$ for a non-duplicate $\bar{y}_j$, at least one of the following statements must hold: $\sum_{i \leq j-1}(\bar{y}_i - y_i) \neq 0$ or $\sum_{i \leq j}(\bar{y}_i - y_i) \neq 0$, meaning that $j - 1 \in \mathcal{J}$ or $j \in \mathcal{J}$.

3. Suppose $\bar{y}_j$ is duplicate and $\sum_{i:\bar{y}_i = \bar{y}_j}(\bar{y}_i - y_i) \neq 0$. Let $k = \min\{i \mid \bar{y}_i = \bar{y}_j\}$ and $l = \max\{i \mid \bar{y}_i = \bar{y}_j\}$. Then at least one of the following statements must hold: $\sum_{i \leq k-1}(\bar{y}_i - y_i) \neq 0$ or $\sum_{i \leq l}(\bar{y}_i - y_i) \neq 0$. If $\sum_{i \leq k-1}(\bar{y}_i - y_i) \neq 0$, we can also see that $\bar{y}_{k-1} < \bar{y}_k$, so $k - 1 \in \mathcal{J}$. Similarly, if $\sum_{i \leq l}(\bar{y}_i - y_i) \neq 0$, then $l \in \mathcal{J}$.

4. Since $\sum_{i:\bar{y}_i = \bar{y}_j}(\bar{y}_i - y_i) = 0$ holds for any duplicate $\bar{y}_j$, if $\bar{y}_i \neq y_i$ holds for one $i$ then there must be at least two of them that satisfies $\bar{y}_i \neq y_i$. If this doesn't hold for all duplicate $\bar{y}_i$, with Part 2 this means that $\bar{y}_j = y_j$ holds for all $j$. This violates our assumption that linear models cannot perfectly fit $Y$.

$\square$

From Lemma A.1.4, we saw that there is a duplicate value of $\bar{y}_j$ such that some of the data points $i$ satisfy $\bar{y}_i = \bar{y}_j$ and $\bar{y}_i \neq y_i$. The proof strategy in this case is essentially the same, but the difference is that we choose one of such duplicate $\bar{y}_j$, and then choose a vector $v \in \mathbb{R}^{d_x}$ to "perturb" the linear least squares solution $[\bar{W}]_{[d_x]}$ in order to break the tie between $i$'s that satisfies $\bar{y}_i = \bar{y}_j$ and $\bar{y}_i \neq y_i$.

We start by defining the minimum among such duplicate values $\bar{y}^*$ of $\bar{y}_j$'s, and a set of indices $j$ that satisfies $\bar{y}_j = \bar{y}^*$.

$$\bar{y}^* = \min\{\bar{y}_j \mid \exists i \neq j \text{ such that } \bar{y}_i = \bar{y}_j \text{ and } \bar{y}_i \neq y_i\},$$

$$\mathcal{J}^* = \{j \in [m] \mid \bar{y}_j = \bar{y}^*\}.$$

Then, we define a subset of $\mathcal{J}^*$:

$$\mathcal{J}^*_{\neq} = \{j \in \mathcal{J}^* \mid \bar{y}_j \neq y_j\}.$$

By Lemma A.1.4, cardinality of $\mathcal{J}^*_{\neq}$ is at least two. Then, we define a special index in $\mathcal{J}^*_{\neq}$:

$$j_1 = \operatorname*{argmax}_{j \in \mathcal{J}^*_{\neq}} \|x_j\|_2,$$

Index $j_1$ is the index of the "longest" $x_j$ among elements in $\mathcal{J}^*_{\neq}$. Using the definition of $j_1$, we can partition $\mathcal{J}^*$ into two sets:

$$\mathcal{J}^*_{\geq} = \{j \in \mathcal{J}^* \mid \langle x_j, x_{j_1} \rangle \geq \|x_{j_1}\|_2^2\}, \ \mathcal{J}^*_{<} = \{j \in \mathcal{J}^* \mid \langle x_j, x_{j_1} \rangle < \|x_{j_1}\|_2^2\}.$$

For the indices in $\mathcal{J}^*$, we can always switch the indices without loss of generality. So we can assume that $j \leq j_1 = \max \mathcal{J}^*_{\geq}$ for all $j \in \mathcal{J}^*_{\geq}$ and $j > j_1$ for all $j \in \mathcal{J}^*_{<}$.

We now define a vector that will be used as the "perturbation" to $[\bar{W}]_{[d_x]}$. Define a vector $v \in \mathbb{R}^{d_x}$, which is a scaled version of $x_{j_1}$:

$$v = \frac{g}{M \|x_{j_1}\|_2} x_{j_1},$$

where the constants $g$ and $M$ are defined to be

$$g = \frac{1}{4} \min \{|\bar{y}_i - \bar{y}_j| \mid i, j \in [m], \bar{y}_i \neq \bar{y}_j\}, \ M = \max_{i \in [m]} \|x_i\|_2.$$

The constant $M$ is the largest $\|x_i\|_2$ among all the indices, and $g$ is one fourth times the minimum gap between all distinct values of $\bar{y}_i$.

Now, consider perturbing $[\bar{W}]_{[d_x]}$ by a vector $-\alpha v^T$. where $\alpha \in (0, 1]$ will be specified later. Observe that

$$\left(\bar{W} - \begin{bmatrix} \alpha v^T & 0 \end{bmatrix}\right) \begin{bmatrix} x_i \\ 1 \end{bmatrix} = \bar{W} \begin{bmatrix} x_i \\ 1 \end{bmatrix} - \alpha v^T x_i = \bar{y}_i - \alpha v^T x_i.$$

Recall that $j \leq j_1 = \max \mathcal{J}^*_{\geq}$ for all $j \in \mathcal{J}^*_{\geq}$ and $j > j_1$ for all $j \in \mathcal{J}^*_{<}$. We are now ready to present the following lemma:

**Lemma A.2.** *Define*

$$j_2 = \operatorname*{argmax}_{j \in \mathcal{J}^*_{<}} \langle x_j, x_{j_1} \rangle, \ \beta = \bar{y}^* - \frac{\alpha}{2} v^T (x_{j_1} + x_{j_2}).$$

*Then,*

$$\bar{y}_i - \alpha v^T x_i - \beta < 0 \text{ for all } i \leq j_1,$$
$$\bar{y}_i - \alpha v^T x_i - \beta > 0 \text{ for all } i > j_1.$$

*Also,* $\sum_{i > j_1} (\bar{y}_i - y_i) - \sum_{i \leq j_1} (\bar{y}_i - y_i) = -2(\bar{y}_{j_1} - y_{j_1}) \neq 0$.

**Proof**  First observe that, for any $x_i$, $|\alpha v^T x_i| \leq \alpha \|v\|_2 \|x_i\|_2 \leq \frac{g}{M} \|x_i\|_2 \leq g$. By definition of $g$, we have $2g < \bar{y}_j - \bar{y}_i$ for any $\bar{y}_i < \bar{y}_j$. Using this, we can see that

$$\bar{y}_i < \bar{y}_j \implies \bar{y}_i - \alpha v^T x_i \leq \bar{y}_i + g < \bar{y}_j - g \leq \bar{y}_j - \alpha v^T x_j. \tag{A.1}$$

In words, if $\bar{y}_i$ and $\bar{y}_j$ are distinct and there is an order $\bar{y}_i < \bar{y}_j$, perturbation of $[\bar{W}]_{[d_x]}$ by $-\alpha v^T$ does not change the order. Also, since $v$ is only a scaled version of $x_{j_1}$, from the definitions of $\mathcal{J}^*_{\geq}$ and $\mathcal{J}^*_{<}$,

$$v^T(x_j - x_{j_1}) \geq 0 \text{ for } j \in \mathcal{J}^*_{\geq} \text{ and } v^T(x_j - x_{j_1}) < 0 \text{ for } j \in \mathcal{J}^*_{<}. \tag{A.2}$$

By definition of $j_2$,

$$v^T(x_{j_2} - x_{j_1}) < 0 \text{ and } v^T(x_{j_2} - x_j) \geq 0 \text{ for all } j \in \mathcal{J}^*_{<}. \tag{A.3}$$

It is left to prove the statement of the lemma using case analysis, using the inequalities (A.1), (A.2), and (A.3). For all $i$'s such that $\bar{y}_i < \bar{y}^* = \bar{y}_{j_1}$,

$$\bar{y}_i - \alpha v^T x_i - \beta = \bar{y}_i - \alpha v^T x_i - \bar{y}^* + \frac{\alpha}{2} v^T (x_{j_1} + x_{j_2})$$
$$= (\bar{y}_i - \alpha v^T x_i) - (\bar{y}^* - \alpha v^T x_{j_1}) + \frac{\alpha}{2} v^T (x_{j_2} - x_{j_1}) < 0.$$

Similarly, for all $i$ such that $\bar{y}_i > \bar{y}^* = \bar{y}_{j_2}$,

$$\bar{y}_i - \alpha v^T x_i - \beta = (\bar{y}_i - \alpha v^T x_i) - (\bar{y}^* - \alpha v^T x_{j_2}) + \frac{\alpha}{2} v^T (x_{j_1} - x_{j_2}) > 0.$$

For $j \in \mathcal{J}_{\geq}^* \ (j \leq j_1)$, we know $\bar{y}_j = \bar{y}^*$, so

$$\bar{y}_j - \alpha v^T x_j - \beta = \left(\bar{y}^* - \alpha v^T x_j\right) - \left(\bar{y}^* - \frac{\alpha}{2} v^T (x_{j_1} + x_{j_2})\right)$$
$$= \alpha v^T [(x_{j_1} - x_j)] + \frac{\alpha}{2} v^T [(x_{j_2} - x_{j_1})] < 0.$$

Also, for $j \in \mathcal{J}_{<}^* \ (j > j_1)$,

$$\bar{y}_j - \alpha v^T x_j - \beta = \left(\bar{y}^* - \alpha v^T x_j\right) - \left(\bar{y}^* - \frac{\alpha}{2} v^T (x_{j_1} + x_{j_2})\right)$$
$$= \frac{\alpha}{2} v^T [(x_{j_1} - x_j) + (x_{j_2} - x_j)] > 0.$$

This finishes the case analysis and proves the first statements of the lemma.

One last thing to prove is that $\sum_{i>j_1} (\bar{y}_i - y_i) - \sum_{i \leq j_1} (\bar{y}_i - y_i) = -2(\bar{y}_{j_1} - y_{j_1}) \neq 0$. Recall from Lemma A.1.2 that for non-duplicate $\bar{y}_j$, we have $\bar{y}_j = y_j$. Also by Lemma A.1.3 if $\bar{y}_j$ is duplicate, $\sum_{i:\bar{y}_i = \bar{y}_j} (\bar{y}_i - y_i) = 0$. So,

$$\sum_{i>j_1} (\bar{y}_i - y_i) - \sum_{i \leq j_1} (\bar{y}_i - y_i) = \sum_{i \in \mathcal{J}_{<}^*} (\bar{y}_i - y_i) - \sum_{i \in \mathcal{J}_{\geq}^*} (\bar{y}_i - y_i).$$

Recall the definition of $\mathcal{J}_{\neq}^* = \{j \in \mathcal{J}^* \mid \bar{y}_j \neq y_j\}$. For $j \in \mathcal{J}^* \backslash \mathcal{J}_{\neq}^*$, $\bar{y}_j = y_j$. So,

$$\sum_{i \in \mathcal{J}_{<}^*} (\bar{y}_i - y_i) - \sum_{i \in \mathcal{J}_{\geq}^*} (\bar{y}_i - y_i) = \sum_{i \in \mathcal{J}_{<}^* \cap \mathcal{J}_{\neq}^*} (\bar{y}_i - y_i) - \sum_{i \in \mathcal{J}_{\geq}^* \cap \mathcal{J}_{\neq}^*} (\bar{y}_i - y_i).$$

Recall the definition of $j_1 = \operatorname{argmax}_{j \in \mathcal{J}_{\neq}^*} \|x_j\|_2$. For any other $j \in \mathcal{J}_{\neq}^* \backslash \{j_1\}$,

$$\|x_{j_1}\|_2^2 \geq \|x_j\|_2 \|x_{j_1}\|_2 \geq \langle x_j, x_{j_1} \rangle,$$

where the first $\geq$ sign is due to definition of $j_1$, and the second is from Cauchy-Schwarz inequality. Since $x_{j_1}$ and $x_j$ are distinct by assumption, they must differ in either length or direction, or both. So, we can check that at least one of "$\geq$" must be strict inequality, so $\|x_{j_1}\|_2^2 > \langle x_j, x_{j_1} \rangle$ for all $j \in \mathcal{J}_{\neq}^* \backslash \{j_1\}$. Thus,

$$\mathcal{J}_{\neq}^* \backslash \{j_1\} = \mathcal{J}_{<}^* \cap \mathcal{J}_{\neq}^* \text{ and } \{j_1\} = \mathcal{J}_{\geq}^* \cap \mathcal{J}_{\neq}^*,$$

proving that

$$\sum_{i>j_1} (\bar{y}_i - y_i) - \sum_{i \leq j_1} (\bar{y}_i - y_i) = \sum_{j \in \mathcal{J}_{\neq}^* \backslash \{j_1\}} (\bar{y}_i - y_i) - (\bar{y}_{j_1} - y_{j_1}).$$

Also, by Lemma A.1.3,

$$0 = \sum_{i \in \mathcal{J}^*} (\bar{y}_i - y_i) = \sum_{i \in \mathcal{J}_{\neq}^*} (\bar{y}_i - y_i) = (\bar{y}_{j_1} - y_{j_1}) + \sum_{j \in \mathcal{J}_{\neq}^* \backslash \{j_1\}} (\bar{y}_i - y_i).$$

Wrapping up all the equalities, we can conclude that

$$\sum_{i>j_1} (\bar{y}_i - y_i) - \sum_{i \leq j_1} (\bar{y}_i - y_i) = -2 (\bar{y}_{j_1} - y_{j_1}),$$

finishing the proof of the last statement. □

It is time to present the parameters $(\tilde{W}_j, \tilde{b}_j)_{j=1}^2$, whose empirical risk is strictly smaller than the local minimum $(\hat{W}_j, \hat{b}_j)_{j=1}^2$ with a sufficiently small choice of $\alpha \in (0, 1]$. Now, let $\gamma$ be a constant such that

$$\gamma = \text{sign}((\bar{y}_{j_1} - y_{j_1})(s_+ - s_-))\frac{\alpha v^T (x_{j_1} - x_{j_2})}{4}. \tag{A.4}$$

Its absolute value is proportional to $\alpha \in (0, 1]$, which is a undetermined number that will be specified at the end of the proof. Since $|\gamma|$ is small enough, we can check that

$$\text{sign}(\bar{y}_i - \alpha v^T x_i - \beta) = \text{sign}(\bar{y}_i - \alpha v^T x_i - \beta + \gamma) = \text{sign}(\bar{y}_i - \alpha v^T x_i - \beta - \gamma).$$

Then, assign parameter values

$$\tilde{W}_1 = \begin{bmatrix} [\bar{W}]_{[d_x]} - \alpha v^T \\ -[\bar{W}]_{[d_x]} + \alpha v^T \\ \mathbf{0}_{(d_1-2) \times d_x} \end{bmatrix}, \; \tilde{b}_1 = \begin{bmatrix} [\bar{W}]_{d_x+1} - \beta + \gamma \\ -[\bar{W}]_{d_x+1} + \beta + \gamma \\ \mathbf{0}_{d_1-2} \end{bmatrix},$$

$$\tilde{W}_2 = \frac{1}{s_+ + s_-} \begin{bmatrix} 1 & -1 & \mathbf{0}_{d_1-2}^T \end{bmatrix}, \; \tilde{b}_2 = \beta.$$

With these parameter values,

$$\tilde{W}_1 x_i + \tilde{b}_1 = \begin{bmatrix} \bar{y}_i - \alpha v^T x_i - \beta + \gamma \\ -\bar{y}_i + \alpha v^T x_i + \beta + \gamma \\ \mathbf{0}_{d_1-2} \end{bmatrix}.$$

As we saw in Lemma A.2, for $i \leq j_1$, $\bar{y}_i - \alpha v^T x_i - \beta + \gamma < 0$ and $-\bar{y}_i + \alpha v^T x_i + \beta + \gamma > 0$. So

$$\hat{y}_i = \tilde{W}_2 \bar{h}_{s_+,s_-}(\tilde{W}_1 x_i + \tilde{b}_1) + \tilde{b}_2$$
$$= \frac{1}{s_+ + s_-} s_-(\bar{y}_i - \alpha v^T x_i - \beta + \gamma) - \frac{1}{s_+ + s_-} s_+(-\bar{y}_i + \alpha v^T x_i + \beta + \gamma) + \beta$$
$$= \bar{y}_i - \alpha v^T x_i - \frac{s_+ - s_-}{s_+ + s_-}\gamma.$$

Similarly, for $i > j_1$, $\bar{y}_i - \alpha v^T x_i - \beta + \gamma > 0$ and $-\bar{y}_i + \alpha v^T x_i + \beta + \gamma < 0$, so

$$\hat{y}_i = \tilde{W}_2 \bar{h}_{s_+,s_-}(\tilde{W}_1 x_i + \tilde{b}_1) + \tilde{b}_2 = \bar{y}_i - \alpha v^T x_i + \frac{s_+ - s_-}{s_+ + s_-}\gamma.$$

Now, the squared error loss of this point is

$$\ell((\tilde{W}_j, \tilde{b}_j)_{j=1}^2) = \frac{1}{2}\|\hat{Y} - Y\|_F^2$$

$$= \frac{1}{2}\sum_{i \leq j_1}\left(\bar{y}_i - \alpha v^T x_i - \frac{s_+ - s_-}{s_+ + s_-}\gamma - y_i\right)^2 + \frac{1}{2}\sum_{i > j_1}\left(\bar{y}_i - \alpha v^T x_i + \frac{s_+ - s_-}{s_+ + s_-}\gamma - y_i\right)^2$$

$$= \frac{1}{2}\sum_{i=1}^m \left(\bar{y}_i - \alpha v^T x_i - y_i\right)^2 + \left[\sum_{i > j_1}\left(\bar{y}_i - \alpha v^T x_i - y_i\right) - \sum_{i \leq j_1}\left(\bar{y}_i - \alpha v^T x_i - y_i\right)\right]\frac{s_+ - s_-}{s_+ + s_-}\gamma + O(\gamma^2)$$

$$= \ell_0(\bar{W}) - \alpha\left[\sum_{i=1}^m (\bar{y}_i - y_i) x_i^T\right]v + O(\alpha^2) + \left[\sum_{i > j_1}(\bar{y}_i - y_i) - \sum_{i \leq j_1}(\bar{y}_i - y_i)\right]\frac{s_+ - s_-}{s_+ + s_-}\gamma + O(\alpha\gamma) + O(\gamma^2).$$

Recall that $\sum_{i=1}^m (\bar{y}_i - y_i) x_i^T = 0$ for least squares estimates $\bar{y}_i$. From Lemma A.2, we saw that $\sum_{i > j_1}(\bar{y}_i - y_i) - \sum_{i \leq j_1}(\bar{y}_i - y_i) = -2(\bar{y}_{j_1} - y_{j_1})$. As seen in the definition of $\gamma$ (A.4), the magnitude of $\gamma$ is proportional to $\alpha$. Substituting (A.4), we can express the loss as

$$\ell((\tilde{W}_j, \tilde{b}_j)_{j=1}^2) = \ell_0(\bar{W}) - \frac{\alpha|(\bar{y}_{j_1} - y_{j_1})(s_+ - s_-)|v^T(x_{j_1} - x_{j_2})}{2(s_+ + s_-)} + O(\alpha^2).$$

Recall that $v^T(x_{j_1} - x_{j_2}) > 0$ from (A.3). Then, for sufficiently small $\alpha \in (0, 1]$,

$$\ell((\tilde{W}_j, \tilde{b}_j)_{j=1}^2) < \ell_0(\bar{W}) = \ell((\hat{W}_j, \hat{b}_j)_{j=1}^2),$$

therefore proving that $(\hat{W}_j, \hat{b}_j)_{j=1}^2$ is a spurious local minimum.

## A3 PROOF OF THEOREM 2

### A3.1 PROOF OF PART 1

Given $v_1, v_2, v_3, v_4 \in \mathbb{R}$ satisfying conditions (C2.1) and (C2.2), we can pick parameter values $(\tilde{W}_j, \tilde{b}_j)_{j=1}^2$ to perfectly fit the given dataset:

$$\tilde{W}_1 = \begin{bmatrix} v_1 & v_2 \\ v_3 & v_4 \end{bmatrix}, \ \tilde{b}_1 = \begin{bmatrix} 0 \\ 0 \end{bmatrix}, \ \tilde{W}_2 = \left( h(v_3)h\left(\tfrac{v_1+v_2}{2}\right) - h(v_1)h\left(\tfrac{v_3+v_4}{2}\right) \right)^{-1} [h(v_3) - h(v_1)], \ \tilde{b}_2 = 0.$$

With these values, we can check that $\hat{Y} = \begin{bmatrix} 0 & 0 & 1 \end{bmatrix}$, hence perfectly fitting $Y$, thus the loss $\ell((\tilde{W}_j, \tilde{b}_j)_{j=1}^2) = 0$.

### A3.2 PROOF OF PART 2

Given conditions (C2.3)–(C2.7) on $v_1, v_2, u_1, u_2 \in \mathbb{R}$, we prove below that there exists a local minimum $(\hat{W}_j, \hat{b}_j)_{j=1}^2$ for which the output of the network is the same as linear least squares model, and its empirical risk is $\ell((\hat{W}_j, \hat{b}_j)_{j=1}^2) = \tfrac{1}{3}$. If the conditions of Part 1 also hold, this local minimum is strictly inferior to the global one.

First, compute the output $\bar{Y}$ of linear least squares model to obtain $\bar{Y} = \begin{bmatrix} \tfrac{1}{3} & \tfrac{1}{3} & \tfrac{1}{3} \end{bmatrix}$. Now assign parameter values

$$\hat{W}_1 = \begin{bmatrix} v_1 & v_1 \\ v_2 & v_2 \end{bmatrix}, \ \hat{b}_1 = \begin{bmatrix} 0 \\ 0 \end{bmatrix}, \ \hat{W}_2 = \begin{bmatrix} u_1 & u_2 \end{bmatrix}, \ \hat{b}_2 = 0.$$

With these values we can check that $\hat{Y} = \begin{bmatrix} \tfrac{1}{3} & \tfrac{1}{3} & \tfrac{1}{3} \end{bmatrix}$, under condition (C2.3): $u_1 h(v_1) + u_2 h(v_2) = \tfrac{1}{3}$. The empirical risk is $\ell((\hat{W}_j, \hat{b}_j)_{j=1}^2) = \tfrac{1}{2}(\tfrac{1}{9} + \tfrac{1}{9} + \tfrac{4}{9}) = \tfrac{1}{3}$.

It remains to show that this is indeed a local minimum of $\ell$. To show this, we apply perturbations to the parameters to see if the risk after perturbation is greater than or equal to $\ell((\hat{W}_j, \hat{b}_j)_{j=1}^2)$. Let the perturbed parameters be

$$\check{W}_1 = \begin{bmatrix} v_1 + \delta_{11} & v_1 + \delta_{12} \\ v_2 + \delta_{21} & v_2 + \delta_{22} \end{bmatrix}, \ \check{b}_1 = \begin{bmatrix} \beta_1 \\ \beta_2 \end{bmatrix}, \ \check{W}_2 = \begin{bmatrix} u_1 + \epsilon_1 & u_2 + \epsilon_2 \end{bmatrix}, \ \check{b}_2 = \gamma, \tag{A.5}$$

where $\delta_{11}, \delta_{12}, \delta_{21}, \delta_{22}, \beta_1, \beta_2, \epsilon_1, \epsilon_2$, and $\gamma$ are small real numbers. The next lemma rearranges the terms in $\ell((\check{W}_j, \check{b}_j)_{j=1}^2)$ into a form that helps us prove local minimality of $(\hat{W}_j, \hat{b}_j)_{j=1}^2$. Appendix A4 gives the proof of Lemma A.3, which includes as a byproduct some equalities on polynomials that may be of wider interest.

**Lemma A.3.** *Assume there exist real numbers $v_1, v_2, u_1, u_2$ such that conditions (C2.3)–(C2.5) hold. Then, for perturbed parameters $(\check{W}_j, \check{b}_j)_{j=1}^2$ defined in (A.5),*

$$\ell((\check{W}_j, \check{b}_j)_{j=1}^2) \geq \tfrac{1}{3} + \alpha_1(\delta_{11} - \delta_{12})^2 + \alpha_2(\delta_{21} - \delta_{22})^2 + \alpha_3(\delta_{11} - \delta_{12})(\delta_{21} - \delta_{22}), \tag{A.6}$$

*where $\alpha_i = \frac{u_i h''(v_i)}{12} + \frac{u_i^2 (h'(v_i))^2}{4} + o(1)$, for $i = 1, 2$, and $\alpha_3 = \frac{u_1 u_2 h'(v_1) h'(v_2)}{2} + o(1)$, and $o(1)$ contains terms that diminish to zero as perturbations vanish.*

To make the the sum of the last three terms of (A.6) nonnegative, we need to satisfy $\alpha_1 \geq 0$ and $\alpha_3^2 - 4\alpha_1\alpha_2 \leq 0$; these inequalities are satisfied for small enough perturbations because of conditions (C2.6)–(C2.7). Thus, we conclude that $\ell((\check{W}_j, \check{b}_j)_{j=1}^2) \geq \tfrac{1}{3} = \ell((\hat{W}_j, \hat{b}_j)_{j=1}^2)$ for small enough perturbations, proving that $(\hat{W}_j, \hat{b}_j)_{j=1}^2$ is a local minimum.

## A4 PROOF OF LEMMA A.3

The goal of this lemma is to prove that

$$\ell((\check{W}_j, \check{b}_j)_{j=1}^2) = \frac{1}{3} + \frac{3}{2}(\text{perturbations})^2 + \left( \frac{u_1 h''(v_1)}{12} + \frac{u_1^2 (h'(v_1))^2}{4} + o(1) \right)(\delta_{11} - \delta_{12})^2$$

$$
+ \left( \frac{u_2 h''(v_2)}{12} + \frac{u_2^2 (h'(v_2))^2}{4} + o(1) \right) (\delta_{21} - \delta_{22})^2
$$
$$
+ \left( \frac{u_1 u_2 h'(v_1) h'(v_2)}{2} + o(1) \right) (\delta_{11} - \delta_{12})(\delta_{21} - \delta_{22}), \tag{A.7}
$$

where $o(1)$ contains terms that diminish to zero as perturbations decrease.

Using the perturbed parameters,

$$
\check{W}_1 X + \check{b}_1 \mathbf{1}_m^T = \begin{bmatrix} v_1 + \delta_{11} + \beta_1 & v_1 + \delta_{12} + \beta_1 & v_1 + \frac{\delta_{11} + \delta_{12}}{2} + \beta_1 \\ v_2 + \delta_{21} + \beta_2 & v_2 + \delta_{22} + \beta_2 & v_2 + \frac{\delta_{21} + \delta_{22}}{2} + \beta_2 \end{bmatrix},
$$

so the empirical risk can be expressed as

$$
\begin{aligned}
& \ell((\check{W}_j, \check{b}_j)_{j=1}^2) \\
=& \frac{1}{2} \| \check{W}_2 h \left( \check{W}_1 X + \check{b}_1 \mathbf{1}_m^T \right) + \check{b}_2 \mathbf{1}_m^T - Y \|_{\mathrm{F}}^2 \\
=& \frac{1}{2} \left[ (u_1 + \epsilon_1) h(v_1 + \delta_{11} + \beta_1) + (u_2 + \epsilon_2) h(v_2 + \delta_{21} + \beta_2) + \gamma \right]^2 \\
& + \frac{1}{2} \left[ (u_1 + \epsilon_1) h(v_1 + \delta_{12} + \beta_1) + (u_2 + \epsilon_2) h(v_2 + \delta_{22} + \beta_2) + \gamma \right]^2 \\
& + \frac{1}{2} \left[ (u_1 + \epsilon_1) h \left( v_1 + \frac{\delta_{11} + \delta_{12}}{2} + \beta_1 \right) + (u_2 + \epsilon_2) h \left( v_2 + \frac{\delta_{21} + \delta_{22}}{2} + \beta_2 \right) + \gamma - 1 \right]^2
\end{aligned} \tag{A.8}
$$

So, the empirical risk (A.8) consists of three terms, one for each training example. By expanding the activation function $h$ using Taylor series expansion and doing algebraic manipulations, we will derive the equation (A.7) from (A.8).

Using the Taylor series expansion, we can express $h(v_1 + \delta_{11} + \beta_1)$ as

$$
h(v_1 + \delta_{11} + \beta_1) = h(v_1) + \sum_{n=1}^{\infty} \frac{h^{(n)}(v_1)}{n!} (\delta_{11} + \beta_1)^n.
$$

Using a similar expansion for $h(v_2 + \delta_{21} + \beta_2)$, the first term of (A.8) can be written as

$$
\begin{aligned}
& \frac{1}{2} \left[ (u_1 + \epsilon_1) h(v_1 + \delta_{11} + \beta_1) + (u_2 + \epsilon_2) h(v_2 + \delta_{21} + \beta_2) + \gamma \right]^2 \\
=& \frac{1}{2} \left[ (u_1 + \epsilon_1) \left( h(v_1) + \sum_{n=1}^{\infty} \frac{h^{(n)}(v_1)}{n!} (\delta_{11} + \beta_1)^n \right) + (u_2 + \epsilon_2) \left( h(v_2) + \sum_{n=1}^{\infty} \frac{h^{(n)}(v_2)}{n!} (\delta_{21} + \beta_2)^n \right) + \gamma \right]^2 \\
=& \frac{1}{2} \left[ \frac{1}{3} + \epsilon_1 h(v_1) + (u_1 + \epsilon_1) \sum_{n=1}^{\infty} \frac{h^{(n)}(v_1)}{n!} (\delta_{11} + \beta_1)^n + \epsilon_2 h(v_2) + (u_2 + \epsilon_2) \sum_{n=1}^{\infty} \frac{h^{(n)}(v_2)}{n!} (\delta_{21} + \beta_2)^n + \gamma \right]^2,
\end{aligned}
$$

where we used $u_1 h(v_1) + u_2 h(v_2) = \frac{1}{3}$. To simplify notation, let us introduce the following function:

$$
t(\delta_1, \delta_2) = \epsilon_1 h(v_1) + \epsilon_2 h(v_2) + \gamma + (u_1 + \epsilon_1) \sum_{n=1}^{\infty} \frac{h^{(n)}(v_1)}{n!} (\delta_1 + \beta_1)^n + (u_2 + \epsilon_2) \sum_{n=1}^{\infty} \frac{h^{(n)}(v_2)}{n!} (\delta_2 + \beta_2)^n.
$$

With this new notation $t(\delta_1, \delta_2)$, after doing similar expansions to the other terms of (A.8), we get

$$
\begin{aligned}
& \ell((\check{W}_j, \check{b}_j)_{j=1}^2) \\
=& \frac{1}{2} \left[ \frac{1}{3} + t(\delta_{11}, \delta_{21}) \right]^2 + \frac{1}{2} \left[ \frac{1}{3} + t(\delta_{12}, \delta_{22}) \right]^2 + \frac{1}{2} \left[ -\frac{2}{3} + t \left( \frac{\delta_{11} + \delta_{12}}{2}, \frac{\delta_{21} + \delta_{22}}{2} \right) \right]^2 \\
=& \frac{1}{3} + \frac{1}{3} \left[ t(\delta_{11}, \delta_{21}) + t(\delta_{12}, \delta_{22}) - 2t \left( \frac{\delta_{11} + \delta_{12}}{2}, \frac{\delta_{21} + \delta_{22}}{2} \right) \right] \\
& + \frac{1}{2} \left[ t(\delta_{11}, \delta_{21}) \right]^2 + \frac{1}{2} \left[ t(\delta_{12}, \delta_{22}) \right]^2 + \frac{1}{2} \left[ t \left( \frac{\delta_{11} + \delta_{12}}{2}, \frac{\delta_{21} + \delta_{22}}{2} \right) \right]^2
\end{aligned} \tag{A.9}
$$

Before we show the lower bounds, we first present the following lemmas that will prove useful shortly. These are simple yet interesting lemmas that might be of independent interest.

**Lemma A.4.** *For $n \geq 2$,*

$$a^n + b^n - 2 \left( \frac{a+b}{2} \right)^n = (a-b)^2 p_n(a,b),$$

*where $p_n$ is a polynomial in $a$ and $b$. All terms in $p_n$ have degree exactly $n - 2$. When $n = 2$, $p_2(a,b) = \frac{1}{2}$.*

**Proof**  The exact formula for $p_n(a,b)$ is as the following:

$$p_n(a,b) = \sum_{k=0}^{n-2} \left[ k+1 - 2^{-n+1} \sum_{l=0}^{k} (k+1-l) \binom{n}{l} \right] a^{n-k-2} b^k.$$

Using this, we can check the lemma is correct just by expanding both sides of the equation. The rest of the proof is straightforward but involves some complicated algebra. So, we omit the details for simplicity. $\qquad \square$

**Lemma A.5.** *For $n_1, n_2 \geq 1$,*

$$a^{n_1} c^{n_2} + b^{n_1} d^{n_2} - 2 \left( \frac{a+b}{2} \right)^{n_1} \left( \frac{c+d}{2} \right)^{n_2}$$

$$= (a-b)^2 q_{n_1,n_2}(a,b,d) + (c-d)^2 q_{n_2,n_1}(c,d,b) + (a-b)(c-d) r_{n_1,n_2}(a,b,c,d)$$

*where $q_{n_1,n_2}$ and $r_{n_1,n_2}$ are polynomials in $a$, $b$, $c$ and $d$. All terms in $q_{n_1,n_2}$ and $r_{n_1,n_2}$ have degree exactly $n_1 + n_2 - 2$. When $n_1 = n_2 = 1$, $q_{1,1}(a,b,d) = 0$ and $r_{1,1}(a,b,c,d) = \frac{1}{2}$.*

**Proof**  The exact formulas for $q_{n_1,n_2}(a,b,d)$, $q_{n_2,n_1}(c,d,b)$, and $r_{n_1,n_2}(a,b,c,d)$ are as the following:

$$q_{n_1,n_2}(a,b,d) = \sum_{k_1=0}^{n_1-2} \left[ k_1 + 1 - 2^{-n_1+1} \sum_{l_1=0}^{k_1} (k_1+1-l_1) \binom{n_1}{l_1} \right] a^{n_1-k_1-2} b^{k_1} d^{n_2},$$

$$q_{n_2,n_1}(c,d,b) = \sum_{k_2=0}^{n_2-2} \left[ k_2 + 1 - 2^{-n_2+1} \sum_{l_2=0}^{k_2} (k_2+1-l_2) \binom{n_2}{l_2} \right] b^{n_1} c^{n_2-k_2-2} d^{k_2},$$

$$r_{n_1,n_2}(a,b,c,d) = \sum_{k_1=0}^{n_1-1} \sum_{k_2=0}^{n_2-1} \left[ 1 - 2^{-n_1-n_2+1} \sum_{l_1=0}^{k_1} \sum_{l_2=0}^{k_2} \binom{n_1}{l_1} \binom{n_2}{l_2} \right] a^{n_1-k_1-1} b^{k_1} c^{n_2-k_2-1} d^{k_2}.$$

Similarly, we can check the lemma is correct just by expanding both sides of the equation. The remaining part of the proof is straightforward, so we will omit the details. $\qquad \square$

Using Lemmas A.4 and A.5, we will expand and simplify the "cross terms" part and "squared terms" part of (A.9). For the "cross terms" in (A.9), let us split $t(\delta_1, \delta_2)$ into two functions $t_1$ and $t_2$:

$$t_1(\delta_1, \delta_2) = \epsilon_1 h(v_1) + \epsilon_2 h(v_2) + \gamma + (u_1 + \epsilon_1) h'(v_1)(\delta_1 + \beta_1) + (u_2 + \epsilon_2) h'(v_2)(\delta_2 + \beta_2)$$

$$t_2(\delta_1, \delta_2) = (u_1 + \epsilon_1) \sum_{n=2}^{\infty} \frac{h^{(n)}(v_1)}{n!} (\delta_1 + \beta_1)^n + (u_2 + \epsilon_2) \sum_{n=2}^{\infty} \frac{h^{(n)}(v_2)}{n!} (\delta_2 + \beta_2)^n,$$

so that $t(\delta_1, \delta_2) = t_1(\delta_1, \delta_2) + t_2(\delta_1, \delta_2)$. It is easy to check that

$$t_1(\delta_{11}, \delta_{21}) + t_1(\delta_{12}, \delta_{22}) - 2 t_1 \left( \frac{\delta_{11} + \delta_{12}}{2}, \frac{\delta_{21} + \delta_{22}}{2} \right) = 0.$$

Also, using Lemma A.4, we can see that

$$(\delta_{11} + \beta_1)^n + (\delta_{12} + \beta_1)^n - 2 \left( \frac{\delta_{11} + \delta_{12}}{2} + \beta_1 \right)^n = (\delta_{11} - \delta_{12})^2 p_n(\delta_{11} + \beta_1, \delta_{12} + \beta_1),$$

$$(\delta_{21} + \beta_2)^n + (\delta_{22} + \beta_2)^n - 2\left(\frac{\delta_{21} + \delta_{22}}{2} + \beta_2\right)^n = (\delta_{21} - \delta_{22})^2 p_n(\delta_{21} + \beta_2, \delta_{22} + \beta_2),$$

so

$$t_2(\delta_{11}, \delta_{21}) + t_2(\delta_{12}, \delta_{22}) - 2t_2\left(\frac{\delta_{11} + \delta_{12}}{2}, \frac{\delta_{21} + \delta_{22}}{2}\right)$$

$$= (u_1 + \epsilon_1)(\delta_{11} - \delta_{12})^2 \sum_{n=2}^{\infty} \frac{h^{(n)}(v_1)}{n!} p_n(\delta_{11} + \beta_1, \delta_{12} + \beta_1)$$

$$+ (u_2 + \epsilon_2)(\delta_{21} - \delta_{22})^2 \sum_{n=2}^{\infty} \frac{h^{(n)}(v_2)}{n!} p_n(\delta_{21} + \beta_2, \delta_{22} + \beta_2).$$

Consider the summation

$$\sum_{n=2}^{\infty} \frac{h^{(n)}(v_1)}{n!} p_n(\delta_{11} + \beta_1, \delta_{12} + \beta_1).$$

We assumed that there exists a constant $c > 0$ such that $|h^{(n)}(v_1)| \le c^n n!$. From this, for small enough perturbations $\delta_{11}$, $\delta_{12}$, and $\beta_1$, we can see that the summation converges, and the summands converge to zero as $n$ increases. Because all the terms in $p_n$ $(n \ge 3)$ are of degree at least one, we can thus write

$$\sum_{n=2}^{\infty} \frac{h^{(n)}(v_1)}{n!} p_n(\delta_{11} + \beta_1, \delta_{12} + \beta_1) = \frac{h''(v_1)}{4} + o(1).$$

So, for small enough $\delta_{11}$, $\delta_{12}$, and $\beta_1$, the term $\frac{h''(v_1)}{4}$ dominates the summation. Similarly, as long as $\delta_{21}$, $\delta_{22}$, and $\beta_2$ are small enough, the summation $\sum_{n=2}^{\infty} \frac{h^{(n)}(v_2)}{n!} p_n(\delta_{21} + \beta_2, \delta_{22} + \beta_2)$ is dominated by $\frac{h''(v_2)}{4}$. In conclusion, for small enough perturbations,

$$t(\delta_{11}, \delta_{21}) + t(\delta_{12}, \delta_{22}) - 2t\left(\frac{\delta_{11} + \delta_{12}}{2}, \frac{\delta_{21} + \delta_{22}}{2}\right)$$

$$= t_2(\delta_{11}, \delta_{21}) + t_2(\delta_{12}, \delta_{22}) - 2t_2\left(\frac{\delta_{11} + \delta_{12}}{2}, \frac{\delta_{21} + \delta_{22}}{2}\right)$$

$$= (u_1 + o(1))\left(\frac{h''(v_1)}{4} + o(1)\right)(\delta_{11} - \delta_{12})^2 + (u_2 + o(1))\left(\frac{h''(v_2)}{4} + o(1)\right)(\delta_{21} - \delta_{22})^2$$

$$= \left(\frac{u_1 h''(v_1)}{4} + o(1)\right)(\delta_{11} - \delta_{12})^2 + \left(\frac{u_2 h''(v_2)}{4} + o(1)\right)(\delta_{21} - \delta_{22})^2. \tag{A.10}$$

Now, it is time to take care of the "squared terms." We will express the terms as

$$\frac{1}{2}[t(\delta_{11}, \delta_{21})]^2 + \frac{1}{2}[t(\delta_{12}, \delta_{22})]^2 + \frac{1}{2}\left[t\left(\frac{\delta_{11} + \delta_{12}}{2}, \frac{\delta_{21} + \delta_{22}}{2}\right)\right]^2$$

$$= \frac{3}{2}\left[t\left(\frac{\delta_{11} + \delta_{12}}{2}, \frac{\delta_{21} + \delta_{22}}{2}\right)\right]^2 + \frac{1}{2}[t(\delta_{11}, \delta_{21})]^2 + \frac{1}{2}[t(\delta_{12}, \delta_{22})]^2 - \left[t\left(\frac{\delta_{11} + \delta_{12}}{2}, \frac{\delta_{21} + \delta_{22}}{2}\right)\right]^2, \tag{A.11}$$

and expand and simplify the terms in

$$\frac{1}{2}[t(\delta_{11}, \delta_{21})]^2 + \frac{1}{2}[t(\delta_{12}, \delta_{22})]^2 - \left[t\left(\frac{\delta_{11} + \delta_{12}}{2}, \frac{\delta_{21} + \delta_{22}}{2}\right)\right]^2.$$

This time, we split $t(\delta_1, \delta_2)$ in another way, this time into three parts:

$$t_3 = \epsilon_1 h(v_1) + \epsilon_2 h(v_2) + \gamma,$$

$$t_4(\delta_1) = (u_1 + \epsilon_1) \sum_{n=1}^{\infty} \frac{h^{(n)}(v_1)}{n!}(\delta_1 + \beta_1)^n,$$

$$t_5(\delta_2) = (u_2 + \epsilon_2) \sum_{n=1}^{\infty} \frac{h^{(n)}(v_2)}{n!} (\delta_2 + \beta_2)^n,$$

so that $t(\delta_1, \delta_2) = t_3 + t_4(\delta_1) + t_5(\delta_2)$. With this,

$$\frac{1}{2} \left[ t(\delta_{11}, \delta_{21}) \right]^2 + \frac{1}{2} \left[ t(\delta_{12}, \delta_{22}) \right]^2 - \left[ t \left( \frac{\delta_{11} + \delta_{12}}{2}, \frac{\delta_{21} + \delta_{22}}{2} \right) \right]^2$$

$$= t_3 \left[ t_4(\delta_{11}) + t_4(\delta_{12}) - 2t_4 \left( \frac{\delta_{11} + \delta_{12}}{2} \right) + t_5(\delta_{21}) + t_5(\delta_{22}) - 2t_5 \left( \frac{\delta_{21} + \delta_{22}}{2} \right) \right]$$

$$+ \frac{1}{2} \left[ (t_4(\delta_{11}))^2 + (t_4(\delta_{12}))^2 - 2 \left( t_4 \left( \frac{\delta_{11} + \delta_{12}}{2} \right) \right)^2 \right]$$

$$+ \frac{1}{2} \left[ (t_5(\delta_{21}))^2 + (t_5(\delta_{22}))^2 - 2 \left( t_5 \left( \frac{\delta_{21} + \delta_{22}}{2} \right) \right)^2 \right]$$

$$+ \left[ t_4(\delta_{11})t_5(\delta_{21}) + t_4(\delta_{12})t_5(\delta_{22}) - 2t_4 \left( \frac{\delta_{11} + \delta_{12}}{2} \right) t_5 \left( \frac{\delta_{21} + \delta_{22}}{2} \right) \right]. \qquad \text{(A.12)}$$

We now have to simplify the equation term by term. We first note that

$$t_4(\delta_{11}) + t_4(\delta_{12}) - 2t_4 \left( \frac{\delta_{11} + \delta_{12}}{2} \right) + t_5(\delta_{21}) + t_5(\delta_{22}) - 2t_5 \left( \frac{\delta_{21} + \delta_{22}}{2} \right)$$

$$= t_2(\delta_{11}, \delta_{21}) + t_2(\delta_{12}, \delta_{22}) - 2t_2 \left( \frac{\delta_{11} + \delta_{12}}{2}, \frac{\delta_{21} + \delta_{22}}{2} \right),$$

so

$$t_3 \left[ t_4(\delta_{11}) + t_4(\delta_{12}) - 2t_4 \left( \frac{\delta_{11} + \delta_{12}}{2} \right) + t_5(\delta_{21}) + t_5(\delta_{22}) - 2t_5 \left( \frac{\delta_{21} + \delta_{22}}{2} \right) \right]$$

$$= t_3 \left[ t_2(\delta_{11}, \delta_{21}) + t_2(\delta_{12}, \delta_{22}) - 2t_2 \left( \frac{\delta_{11} + \delta_{12}}{2}, \frac{\delta_{21} + \delta_{22}}{2} \right) \right]$$

$$= o(1) \left[ \left( \frac{u_1 h''(v_1)}{4} + o(1) \right) (\delta_{11} - \delta_{12})^2 + \left( \frac{u_2 h''(v_2)}{4} + o(1) \right) (\delta_{21} - \delta_{22})^2 \right], \qquad \text{(A.13)}$$

as seen in (A.10). Next, we have

$$(t_4(\delta_{11}))^2 + (t_4(\delta_{12}))^2 - 2 \left( t_4 \left( \frac{\delta_{11} + \delta_{12}}{2} \right) \right)^2$$

$$= (u_1 + \epsilon_1)^2 \sum_{n_1, n_2 = 1}^{\infty} \frac{h^{(n_1)}(v_1) h^{(n_2)}(v_1)}{n_1! n_2!} \left[ (\delta_{11} + \beta_1)^{n_1 + n_2} + (\delta_{12} + \beta_1)^{n_1 + n_2} - 2 \left( \frac{\delta_{11} + \delta_{12}}{2} + \beta_1 \right)^{n_1 + n_2} \right],$$

$$= (u_1 + \epsilon_1)^2 (\delta_{11} - \delta_{12})^2 \sum_{n_1, n_2 = 1}^{\infty} \frac{h^{(n_1)}(v_1) h^{(n_2)}(v_1)}{n_1! n_2!} p_{n_1 + n_2}(\delta_{11} + \beta_1, \delta_{12} + \beta_1)$$

$$= \left( \frac{u_1^2 (h'(v_1))^2}{2} + o(1) \right) (\delta_{11} - \delta_{12})^2, \qquad \text{(A.14)}$$

when perturbations are small enough. We again used Lemma A.4 in the second equality sign, and the facts that $p_{n_1 + n_2}(\cdot) = o(1)$ whenever $n_1 + n_2 > 2$ and that $p_2(\cdot) = \frac{1}{2}$. In a similar way,

$$(t_5(\delta_{21}))^2 + (t_5(\delta_{22}))^2 - 2 \left( t_5 \left( \frac{\delta_{21} + \delta_{22}}{2} \right) \right)^2 = \left( \frac{u_2^2 (h'(v_2))^2}{2} + o(1) \right) (\delta_{21} - \delta_{22})^2. \quad \text{(A.15)}$$

Lastly,

$$t_4(\delta_{11})t_5(\delta_{21}) + t_4(\delta_{12})t_5(\delta_{22}) - 2t_4 \left( \frac{\delta_{11} + \delta_{12}}{2} \right) t_5 \left( \frac{\delta_{21} + \delta_{22}}{2} \right)$$

$$= (u_1 + \epsilon_1)(u_2 + \epsilon_2) \sum_{n_1, n_2 = 1}^{\infty} \frac{h^{(n_1)}(v_1) h^{(n_2)}(v_2)}{n_1! n_2!} \left[ (\delta_{11} + \beta_1)^{n_1} (\delta_{21} + \beta_2)^{n_2} \right.$$

$$+ (\delta_{12} + \beta_1)^{n_1}(\delta_{22} + \beta_2)^{n_2} - 2\left(\frac{\delta_{11} + \delta_{12}}{2} + \beta_1\right)^{n_1}\left(\frac{\delta_{21} + \delta_{22}}{2} + \beta_2\right)^{n_2}\Bigg],$$

$$=(u_1 + \epsilon_1)(u_2 + \epsilon_2)\Bigg[(\delta_{11} - \delta_{12})^2 \sum_{n_1,n_2=1}^{\infty} \frac{h^{(n_1)}(v_1)h^{(n_2)}(v_2)}{n_1!n_2!}q_{n_1,n_2}(\delta_{11} + \beta_1, \delta_{12} + \beta_1, \delta_{22} + \beta_2)$$

$$+ (\delta_{21} - \delta_{22})^2 \sum_{n_1,n_2=1}^{\infty} \frac{h^{(n_1)}(v_1)h^{(n_2)}(v_2)}{n_1!n_2!}q_{n_2,n_1}(\delta_{21} + \beta_2, \delta_{22} + \beta_2, \delta_{12} + \beta_1)$$

$$+ (\delta_{11} - \delta_{12})(\delta_{21} - \delta_{22}) \sum_{n_1,n_2=1}^{\infty} \frac{h^{(n_1)}(v_1)h^{(n_2)}(v_2)}{n_1!n_2!}r_{n_1,n_2}(\delta_{11} + \beta_1, \delta_{12} + \beta_1, \delta_{21} + \beta_2, \delta_{22} + \beta_2)\Bigg]$$

$$=(u_1 u_2 + o(1))\left[(\delta_{11} - \delta_{12})^2 o(1) + (\delta_{21} - \delta_{22})^2 o(1) + (\delta_{11} - \delta_{12})(\delta_{21} - \delta_{22})\left(\frac{h'(v_1)h'(v_2)}{2} + o(1)\right)\right],$$
(A.16)

where the second equality sign used Lemma A.5 and the third equality sign used the facts that $q_{n_1,n_2}(\cdot) = o(1)$ and $r_{n_1,n_2}(\cdot) = o(1)$ whenever $n_1 + n_2 > 2$, and that $q_{1,1}(\cdot) = 0$ and $r_{1,1}(\cdot) = \frac{1}{2}$. If we substitute (A.13), (A.14), (A.15), and (A.16) into (A.12),

$$\frac{1}{2}\left[t(\delta_{11}, \delta_{21})\right]^2 + \frac{1}{2}\left[t(\delta_{12}, \delta_{22})\right]^2 - \left[t\left(\frac{\delta_{11} + \delta_{12}}{2}, \frac{\delta_{21} + \delta_{22}}{2}\right)\right]^2$$

$$=o(1)\left[\left(\frac{u_1 h''(v_1)}{4} + o(1)\right)(\delta_{11} - \delta_{12})^2 + \left(\frac{u_2 h''(v_2)}{4} + o(1)\right)(\delta_{21} - \delta_{22})^2\right]$$

$$+ \frac{1}{2}\left(\frac{u_1^2(h'(v_1))^2}{2} + o(1)\right)(\delta_{11} - \delta_{12})^2 + \frac{1}{2}\left(\frac{u_2^2(h'(v_2))^2}{2} + o(1)\right)(\delta_{21} - \delta_{22})^2$$

$$+ (u_1 u_2 + o(1))\left[(\delta_{11} - \delta_{12})^2 o(1) + (\delta_{21} - \delta_{22})^2 o(1) + (\delta_{11} - \delta_{12})(\delta_{21} - \delta_{22})\left(\frac{h'(v_1)h'(v_2)}{2} + o(1)\right)\right]$$

$$=\left(\frac{u_1^2(h'(v_1))^2}{4} + o(1)\right)(\delta_{11} - \delta_{12})^2 + \left(\frac{u_2^2(h'(v_2))^2}{4} + o(1)\right)(\delta_{21} - \delta_{22})^2$$

$$+ \left(\frac{u_1 u_2 h'(v_1)h'(v_2)}{2} + o(1)\right)(\delta_{11} - \delta_{12})(\delta_{21} - \delta_{22}).$$
(A.17)

We are almost done. If we substitute (A.10), (A.11), and (A.17) into (A.9), we can get

$$\ell((\check{W}_j, \check{b}_j)_{j=1}^2)$$

$$=\frac{1}{3} + \frac{3}{2}\left[t\left(\frac{\delta_{11} + \delta_{12}}{2}, \frac{\delta_{21} + \delta_{22}}{2}\right)\right]^2$$

$$+ \left(\frac{u_1 h''(v_1)}{12} + o(1)\right)(\delta_{11} - \delta_{12})^2 + \left(\frac{u_2 h''(v_2)}{12} + o(1)\right)(\delta_{21} - \delta_{22})^2$$

$$+ \left(\frac{u_1^2(h'(v_1))^2}{4} + o(1)\right)(\delta_{11} - \delta_{12})^2 + \left(\frac{u_2^2(h'(v_2))^2}{4} + o(1)\right)(\delta_{21} - \delta_{22})^2$$

$$+ \left(\frac{u_1 u_2 h'(v_1)h'(v_2)}{2} + o(1)\right)(\delta_{11} - \delta_{12})(\delta_{21} - \delta_{22})$$

$$=\frac{1}{3} + \frac{3}{2}\left[t\left(\frac{\delta_{11} + \delta_{12}}{2}, \frac{\delta_{21} + \delta_{22}}{2}\right)\right]^2 + \left(\frac{u_1 h''(v_1)}{12} + \frac{u_1^2(h'(v_1))^2}{4} + o(1)\right)(\delta_{11} - \delta_{12})^2$$

$$+ \left(\frac{u_2 h''(v_2)}{12} + \frac{u_2^2(h'(v_2))^2}{4} + o(1)\right)(\delta_{21} - \delta_{22})^2 + \left(\frac{u_1 u_2 h'(v_1)h'(v_2)}{2} + o(1)\right)(\delta_{11} - \delta_{12})(\delta_{21} - \delta_{22}),$$

which is the equation (A.7) that we were originally aiming to show.

## A5 PROOF OF COROLLARY 3

For the proof of this corollary, we present the values of real numbers that satisfy assumptions (C2.1)–(C2.7), for each activation function listed in the corollary: sigmoid, tanh, arctan, exponential linear units (ELU, Clevert et al. (2015)), scaled exponential linear units (SELU, Klambauer et al. (2017)).

To remind the readers what the assumptions were, we list the assumptions again. For (C2.1)–(C2.2), there exist real numbers $v_1, v_2, v_3, v_4 \in \mathbb{R}$ such that

(C2.1) $h(v_1)h(v_4) = h(v_2)h(v_3)$,

(C2.2) $h(v_1)h\left(\frac{v_3+v_4}{2}\right) \neq h(v_3)h\left(\frac{v_1+v_2}{2}\right)$.

For (C2.3)–(C2.7), there exist real numbers $v_1, v_2, u_1, u_2 \in \mathbb{R}$ such that the following assumptions hold:

(C2.3) $u_1 h(v_1) + u_2 h(v_2) = \frac{1}{3}$,

(C2.4) $h$ is infinitely differentiable at $v_1$ and $v_2$,

(C2.5) There exists a constant $c > 0$ such that $|h^{(n)}(v_1)| \leq c^n n!$ and $|h^{(n)}(v_2)| \leq c^n n!$.

(C2.6) $(u_1 h'(v_1))^2 + \frac{u_1 h''(v_1)}{3} > 0$,

(C2.7) $(u_1 h'(v_1)u_2 h'(v_2))^2 < ((u_1 h'(v_1))^2 + \frac{u_1 h''(v_1)}{3})((u_2 h'(v_2))^2 + \frac{u_2 h''(v_2)}{3})$.

For each function, we now present the appropriate real numbers that satisfy the assumptions.

### A5.1 SIGMOID

When $h$ is sigmoid,

$$h(x) = \frac{1}{1 + \exp(-x)}, \ h^{-1}(x) = \log\left(\frac{x}{1-x}\right).$$

Assumptions (C2.1)–(C2.2) are satisfied by

$$(v_1, v_2, v_3, v_4) = \left(h^{-1}\left(\frac{1}{2}\right), h^{-1}\left(\frac{1}{4}\right), h^{-1}\left(\frac{1}{4}\right), h^{-1}\left(\frac{1}{8}\right)\right),$$

and assumptions (C2.3)–(C2.7) are satisfied by

$$(v_1, v_2, u_1, u_1) = \left(h^{-1}\left(\frac{1}{4}\right), h^{-1}\left(\frac{1}{4}\right), \frac{2}{3}, \frac{2}{3}\right).$$

Among them, (C2.4)–(C2.5) follow because sigmoid function is an real analytic function Krantz & Parks (2002).

### A5.2 TANH

When $h$ is hyperbolic tangent, assumptions (C2.1)–(C2.2) are satisfied by

$$(v_1, v_2, v_3, v_4) = \left(\tanh^{-1}\left(\frac{1}{2}\right), \tanh^{-1}\left(\frac{1}{4}\right), \tanh^{-1}\left(\frac{1}{4}\right), \tanh^{-1}\left(\frac{1}{8}\right)\right),$$

and assumptions (C2.3)–(C2.7) are satisfied by

$$(v_1, v_2, u_1, u_1) = \left(\tanh^{-1}\left(\frac{1}{2}\right), \tanh^{-1}\left(\frac{1}{2}\right), 1, -\frac{1}{3}\right),$$

Assumptions (C2.4)–(C2.5) hold because hyperbolic tangent function is real analytic.

### A5.3 ARCTAN

When $h$ is inverse tangent, assumptions (C2.1)–(C2.2) are satisfied by

$$(v_1, v_2, v_3, v_4) = \left( \tan\left(\frac{1}{2}\right), \tan\left(\frac{1}{4}\right), \tan\left(\frac{1}{4}\right), \tan\left(\frac{1}{8}\right) \right),$$

and assumptions (C2.3)–(C2.7) are satisfied by

$$(v_1, v_2, u_1, u_1) = \left( \tan\left(\frac{1}{2}\right), \tan\left(\frac{1}{2}\right), 1, -\frac{1}{3} \right),$$

Assumptions (C2.4)–(C2.5) hold because inverse tangent function is real analytic.

### A5.4 QUADRATIC

When $h$ is quadratic, assumptions (C2.1)–(C2.2) are satisfied by

$$(v_1, v_2, v_3, v_4) = \left( 1, \frac{1}{2}, \frac{1}{2}, -\frac{1}{4} \right),$$

and assumptions (C2.3)–(C2.7) are satisfied by

$$(v_1, v_2, u_1, u_1) = \left( 1, 1, \frac{1}{6}, \frac{1}{6} \right),$$

Assumptions (C2.4)–(C2.5) hold because quadratic function is real analytic.

### A5.5 ELU AND SELU

When $h$ is ELU or SELU,

$$h(x) = \lambda \begin{cases} x & x \geq 0 \\ \alpha(\exp(x) - 1) & x < 0 \end{cases}, \quad h^{-1}(x) = \begin{cases} x/\lambda & x \geq 0 \\ \log\left(\frac{x}{\lambda\alpha} + 1\right) & x < 0 \end{cases},$$

$$h'(x) = \begin{cases} \lambda & x \geq 0 \\ \lambda\alpha\exp(x) & x < 0 \end{cases}, \quad h''(x) = \begin{cases} 0 & x \geq 0 \\ \lambda\alpha\exp(x) & x < 0 \end{cases},$$

where $\alpha > 0$, and $\lambda = 1$ (ELU) or $\lambda > 1$ (SELU). In this case, assumptions (C2.1)–(C2.2) are satisfied by

$$(v_1, v_2, v_3, v_4) = \left( h^{-1}\left(-\frac{\lambda\alpha}{2}\right), h^{-1}\left(-\frac{\lambda\alpha}{4}\right), h^{-1}\left(-\frac{\lambda\alpha}{4}\right), h^{-1}\left(-\frac{\lambda\alpha}{8}\right) \right).$$

Assumptions (C2.3)–(C2.7) are satisfied by

$$(v_1, v_2, u_1, u_2) = \left( \frac{1}{3}, \log\left(\frac{2}{3}\right), \frac{2}{\lambda}, \frac{1}{\lambda\alpha} \right),$$

where (C2.4)–(C2.5) are satisfied because $h(x)$ is real analytic at $v_1$ and $v_2$.

## A6 PROOF OF THEOREM 2 FOR "RELU-LIKE" ACTIVATION FUNCTIONS.

Recall the piecewise linear nonnegative homogeneous activation function

$$\bar{h}_{s_+, s_-}(x) = \begin{cases} s_+ x & x \geq 0 \\ s_- x & x < 0, \end{cases}$$

where $s_+ > 0$, $s_- \geq 0$ and $s_+ \neq s_-$, we will prove that the statements of Theorem 2 hold for $\bar{h}_{s_+, s_-}$.

### A6.1 PROOF OF PART 1

In the case of $s_- > 0$, assumptions (C2.1)–(C2.2) are satisfied by

$$(v_1, v_2, v_3, v_4) = \left( \frac{1}{s_+}, -\frac{1}{s_-}, -\frac{1}{s_-}, \frac{1}{s_+} \right).$$

The rest of the proof can be done in exactly the same way as the proof of Theorem 2.1, provided in Appendix A3.

For $s_- = 0$, which corresponds to the case of ReLU, define parameters

$$\tilde{W}_1 = \begin{bmatrix} 0 & 2 \\ -2 & 1 \end{bmatrix}, \; \tilde{b}_1 = \begin{bmatrix} 0 \\ 0 \end{bmatrix}, \; \tilde{W}_2 = \begin{bmatrix} \frac{1}{s_+} & -\frac{2}{s_+} \end{bmatrix}, \; \tilde{b}_2 = 0.$$

We can check that

$$\bar{h}_{s_+,s_-}(\tilde{W}_1 X + \tilde{b}_1 \mathbf{1}_3^T) = s_+ \begin{bmatrix} 0 & 2 & 1 \\ 0 & 1 & 0 \end{bmatrix},$$

so

$$\tilde{W}_2 \bar{h}_{s_+,s_-}(\tilde{W}_1 X + \tilde{b}_1 \mathbf{1}_3^T) + \tilde{b}_2 \mathbf{1}_3^T = \begin{bmatrix} 0 & 0 & 1 \end{bmatrix}.$$

### A6.2 PROOF OF PART 2

Assumptions (C2.3)–(C2.6) are satisfied by

$$(v_1, v_2, u_1, u_1) = \left( \frac{1}{4s_+}, \frac{1}{4s_+}, \frac{2}{3}, \frac{2}{3} \right).$$

Assign parameter values

$$\hat{W}_1 = \begin{bmatrix} v_1 & v_1 \\ v_2 & v_2 \end{bmatrix}, \; \hat{b}_1 = \begin{bmatrix} 0 \\ 0 \end{bmatrix}, \; \hat{W}_2 = \begin{bmatrix} u_1 & u_2 \end{bmatrix}, \; \hat{b}_2 = 0.$$

It is easy to compute that the output of the neural network is $\hat{Y} = \begin{bmatrix} \frac{1}{3} & \frac{1}{3} & \frac{1}{3} \end{bmatrix}$, so $\ell((\hat{W}_j, \hat{b}_j)_{j=1}^2) = \frac{1}{3}$.

Now, it remains to show that this is indeed a local minimum of $\ell$. To show this, we apply perturbations to the parameters to see if the risk after perturbation is greater than or equal to $\ell((\hat{W}_j, \hat{b}_j)_{j=1}^2)$. Let the perturbed parameters be

$$\check{W}_1 = \begin{bmatrix} v_1 + \delta_{11} & v_1 + \delta_{12} \\ v_2 + \delta_{21} & v_2 + \delta_{22} \end{bmatrix}, \; \check{b}_1 = \begin{bmatrix} \beta_1 \\ \beta_2 \end{bmatrix}, \; \check{W}_2 = \begin{bmatrix} u_1 + \epsilon_1 & u_2 + \epsilon_2 \end{bmatrix}, \; \check{b}_2 = \gamma,$$

where $\delta_{11}, \delta_{12}, \delta_{21}, \delta_{22}, \beta_1, \beta_2, \epsilon_1, \epsilon_2,$ and $\gamma$ are small enough real numbers.

Using the perturbed parameters,

$$\check{W}_1 X + \check{b}_1 \mathbf{1}_m^T = \begin{bmatrix} v_1 + \delta_{11} + \beta_1 & v_1 + \delta_{12} + \beta_1 & v_1 + \frac{\delta_{11} + \delta_{12}}{2} + \beta_1 \\ v_2 + \delta_{21} + \beta_2 & v_2 + \delta_{22} + \beta_2 & v_2 + \frac{\delta_{21} + \delta_{22}}{2} + \beta_2 \end{bmatrix},$$

so the empirical risk can be expressed as

$$\ell((\check{W}_j, \check{b}_j)_{j=1}^2)$$
$$= \frac{1}{2} \| \check{W}_2 \bar{h}_{s_+,s_-}(\check{W}_1 X + \check{b}_1 \mathbf{1}_m^T) + \check{b}_2 \mathbf{1}_m^T - Y \|_F^2$$
$$= \frac{1}{2} [(u_1 + \epsilon_1) s_+ (v_1 + \delta_{11} + \beta_1) + (u_2 + \epsilon_2) s_+ (v_2 + \delta_{21} + \beta_2) + \gamma]^2$$
$$+ \frac{1}{2} [(u_1 + \epsilon_1) s_+ (v_1 + \delta_{12} + \beta_1) + (u_2 + \epsilon_2) s_+ (v_2 + \delta_{22} + \beta_2) + \gamma]^2$$
$$+ \frac{1}{2} \left[ (u_1 + \epsilon_1) s_+ \left( v_1 + \frac{\delta_{11} + \delta_{12}}{2} + \beta_1 \right) + (u_2 + \epsilon_2) s_+ \left( v_2 + \frac{\delta_{21} + \delta_{22}}{2} + \beta_2 \right) + \gamma - 1 \right]^2.$$

To simplify notation, let us introduce the following function:

$$t(\delta_1, \delta_2) = s_+\epsilon_1 v_1 + s_+\epsilon_2 v_2 + \gamma + s_+(u_1 + \epsilon_1)(\delta_1 + \beta_1) + s_+(u_2 + \epsilon_2)(\delta_2 + \beta_2)$$

It is easy to check that

$$t(\delta_{11}, \delta_{21}) + t(\delta_{12}, \delta_{22}) - 2t\left(\frac{\delta_{11} + \delta_{12}}{2}, \frac{\delta_{21} + \delta_{22}}{2}\right) = 0.$$

With this new notation $t(\delta_1, \delta_2)$, we get

$$\ell((\check{W}_j, \check{b}_j)_{j=1}^2)$$

$$= \frac{1}{2}\left[\frac{1}{3} + t(\delta_{11}, \delta_{21})\right]^2 + \frac{1}{2}\left[\frac{1}{3} + t(\delta_{12}, \delta_{22})\right]^2 + \frac{1}{2}\left[-\frac{2}{3} + t\left(\frac{\delta_{11} + \delta_{12}}{2}, \frac{\delta_{21} + \delta_{22}}{2}\right)\right]^2$$

$$= \frac{1}{3} + \frac{1}{3}\left[t(\delta_{11}, \delta_{21}) + t(\delta_{12}, \delta_{22}) - 2t\left(\frac{\delta_{11} + \delta_{12}}{2}, \frac{\delta_{21} + \delta_{22}}{2}\right)\right]$$

$$+ \frac{1}{2}\left[t(\delta_{11}, \delta_{21})\right]^2 + \frac{1}{2}\left[t(\delta_{12}, \delta_{22})\right]^2 + \frac{1}{2}\left[t\left(\frac{\delta_{11} + \delta_{12}}{2}, \frac{\delta_{21} + \delta_{22}}{2}\right)\right]^2 \geq \frac{1}{3} = \ell((\hat{W}_j, \hat{b}_j)_{j=1}^2).$$

## A7    PROOF OF THEOREM 4

Before we start, note the following partial derivatives, which can be computed using straightforward matrix calculus:

$$\frac{\partial \ell}{\partial W_j} = (W_{H+1:j+1})^T \nabla \ell_0(W_{H+1:1})(W_{j-1:1})^T,$$

for all $j \in [H + 1]$.

### A7.1    PROOF OF PART 1, IF $d_y \geq d_x$

For Part 1, we must show that if $\nabla \ell_0(\hat{W}_{H+1:1}) \neq 0$ then $(\hat{W}_j)_{j=1}^{H+1}$ is a saddle point of $\ell$. Thus, we show that $(\hat{W}_j)_{j=1}^{H+1}$ is neither a local minimum nor a local maximum. More precisely, for each $j$, let $\mathcal{B}_\epsilon(W_j)$ be an $\epsilon$-Frobenius-norm-ball centered at $W_j$, and $\prod_{j=1}^{H+1} \mathcal{B}_\epsilon(W_j)$ their Cartesian product. We wish to show that for every $\epsilon > 0$, there exist tuples $(P_j)_{j=1}^{H+1}, (Q_j)_{j=1}^{H+1} \in \prod_{j=1}^{H+1} \mathcal{B}_\epsilon(\hat{W}_j)$ such that

$$\ell((P_j)_{j=1}^{H+1}) > \ell((\hat{W}_j)_{j=1}^{H+1}) > \ell((Q_j)_{j=1}^{H+1}). \tag{A.18}$$

To prove (A.18), we exploit $\ell((\hat{W}_j)_{j=1}^{H+1}) = \ell_0(\hat{W}_{H+1:1})$, and the assumption $\nabla \ell_0(\hat{W}_{H+1:1}) \neq 0$. The key idea is to perturb the tuple $(\hat{W}_j)_{j=1}^{H+1}$ so that the directional derivative of $\ell_0$ along $P_{H+1:1} - \hat{W}_{H+1:1}$ is positive. Since $\ell_0$ is differentiable, if $P_{H+1:1} - \hat{W}_{H+1:1}$ is small, then

$$\ell((P_j)_{j=1}^{H+1}) = \ell_0(P_{H+1:1}) > \ell_0(\hat{W}_{H+1:1}) = \ell((\hat{W}_j)_{j=1}^{H+1}).$$

Similarly, we can show $\ell((Q_j)_{j=1}^{H+1}) < \ell((\hat{W}_j)_{j=1}^{H+1})$. The key challenge lies in constructing these perturbations; we outline our approach below; this construction may be of independent interest too. For this section, we assume that $d_x \geq d_y$ for simplicity; the case $d_y \geq d_x$ is treated in Appendix A7.2.

Since $\nabla \ell_0(\hat{W}_{H+1:1}) \neq 0$, $\text{col}(\nabla \ell_0(\hat{W}_{H+1:1}))^\perp$ must be a strict subspace of $\mathbb{R}^{d_y}$. Consider $\partial \ell / \partial W_1$ at a critical point to see that $(\hat{W}_{H+1:2})^T \nabla \ell_0(\hat{W}_{H+1:1}) = 0$, so $\text{col}(\hat{W}_{H+1:2}) \subseteq \text{col}(\nabla \ell_0(\hat{W}_{H+1:1}))^\perp \subsetneq \mathbb{R}^{d_y}$. This strict inclusion implies $\text{rank}(\hat{W}_{H+1:2}) < d_y \leq d_1$, so that $\text{null}(\hat{W}_{H+1:2})$ is not a trivial subspace. Moreover, $\text{null}(\hat{W}_{H+1:2}) \supseteq \text{null}(\hat{W}_{H:2}) \supseteq \cdots \supseteq \text{null}(\hat{W}_2)$. We can split the proof into two cases: $\text{null}(\hat{W}_{H+1:2}) \neq \text{null}(\hat{W}_{H:2})$ and $\text{null}(\hat{W}_{H+1:2}) = \text{null}(\hat{W}_{H:2})$.

Let the SVD of $\nabla \ell_0(\hat{W}_{H+1:1}) = U_l \Sigma U_r^T$. Recall $[U_l]_{\cdot,1}$ and $[U_r]_{\cdot,1}$ denote first columns of $U_l$ and $U_r$, respectively.

**Case 1: $\text{null}(\hat{W}_{H+1:2}) \neq \text{null}(\hat{W}_{H:2})$.** In this case, $\text{null}(\hat{W}_{H+1:2}) \supsetneq \text{null}(\hat{W}_{H:2})$. We will perturb $\hat{W}_1$ and $\hat{W}_{H+1}$ to obtain the tuples $(P_j)_{j=1}^{H+1}$ and $(Q_j)_{j=1}^{H+1}$. To create our perturbation, we choose two unit vectors as follows:

$$v_0 = [U_r]_{.,1}, \ v_1 \in \text{null}(\hat{W}_{H+1:2}) \cap \text{null}(\hat{W}_{H:2})^\perp.$$

Then, define $\Delta_1 := \epsilon v_1 v_0^T \in \mathbb{R}^{d_1 \times d_x}$, and $V_1 := \hat{W}_1 + \Delta_1 \in \mathcal{B}_\epsilon(\hat{W}_1)$. Since $v_1$ lies in $\text{null}(\hat{W}_{H+1:2})$, observe that

$$\hat{W}_{H+1:2}V_1 = \hat{W}_{H+1:1} + \epsilon\hat{W}_{H+1:2}v_1 v_0^T = \hat{W}_{H+1:1}.$$

With this definition of $V_1$, we can also see that

$$\nabla\ell_0(\hat{W}_{H+1:1})V_1^T(\hat{W}_{H:2})^T = \nabla\ell_0(\hat{W}_{H+1:1})(\hat{W}_{H:1})^T + \epsilon\nabla\ell_0(\hat{W}_{H+1:1})v_0 v_1^T(\hat{W}_{H:2})^T.$$

Note that $\nabla\ell_0(\hat{W}_{H+1:1})(\hat{W}_{H:1})^T$ is equal to $\partial\ell/\partial W_{H+1}$ at a critical point, hence is zero. Since $v_0 = [U_r]_{.,1}$, we have $\nabla\ell_0(\hat{W}_{H+1:1})v_0 = \sigma_{\max}(\nabla\ell_0(\hat{W}_{H+1:1}))[U_l]_{.,1}$, which is a nonzero column vector, and since $v_1 \in \text{null}(\hat{W}_{H:2})^\perp = \text{row}(\hat{W}_{H:2})$, $v_1^T(\hat{W}_{H:2})^T$ is a nonzero row vector. From this observation, $\nabla\ell_0(\hat{W}_{H+1:1})v_0 v_1^T(\hat{W}_{H:2})^T$ is nonzero, and so is $\nabla\ell_0(\hat{W}_{H+1:1})V_1^T(\hat{W}_{H:2})^T$.

We are now ready to define the perturbation on $\hat{W}_{H+1}$:

$$\Delta_{H+1} := \frac{\epsilon\nabla\ell_0(\hat{W}_{H+1:1})V_1^T(\hat{W}_{H:2})^T}{\|\nabla\ell_0(\hat{W}_{H+1:1})V_1^T(\hat{W}_{H:2})^T\|_{\text{F}}},$$

so that $\hat{W}_{H+1} + \Delta_{H+1} \in B_\epsilon(\hat{W}_{H+1})$. Then, observe that

$$\langle\Delta_{H+1}\hat{W}_{H:2}V_1, \nabla\ell_0(\hat{W}_{H+1:1})\rangle = \langle\Delta_{H+1}, \nabla\ell_0(\hat{W}_{H+1:1})V_1^T(\hat{W}_{H:2})^T\rangle > 0,$$

by definition of $\Delta_{H+1}$. In other words, $\Delta_{H+1}\hat{W}_{H:2}V_1$ is an ascent direction of $\ell_0$ at $\hat{W}_{H+1:1}$. Now choose the tuples

$$(P_j)_{j=1}^{H+1} = (V_1, \hat{W}_2, \ldots, \hat{W}_H, \hat{W}_{H+1} + \eta\Delta_{H+1}),$$
$$(Q_j)_{j=1}^{H+1} = (V_1, \hat{W}_2, \ldots, \hat{W}_H, \hat{W}_{H+1} - \eta\Delta_{H+1}),$$

where $\eta \in (0, 1]$ is chosen suitably. It is easy to verify that $(P_j)_{j=1}^{H+1}, (Q_j)_{j=1}^{H+1} \in \prod_{j=1}^{H+1}\mathcal{B}_\epsilon(\hat{W}_j)$, and that the products

$$P_{H+1:1} = \hat{W}_{H+1:1} + \eta\Delta_{H+1}\hat{W}_{H:2}V_1,$$
$$Q_{H+1:1} = \hat{W}_{H+1:1} - \eta\Delta_{H+1}\hat{W}_{H:2}V_1.$$

Since $\ell_0$ is differentiable, for small enough $\eta \in (0, 1]$, $\ell_0(P_{H+1:1}) > \ell_0(\hat{W}_{H+1:1}) > \ell_0(Q_{H+1:1})$, proving (A.18). This construction is valid for any $\epsilon > 0$, so we are done.

**Case 2: $\text{null}(\hat{W}_{H+1:2}) = \text{null}(\hat{W}_{H:2})$.** By and large, the proof of this case goes the same, except that we need a little more care on what perturbations to make. Define

$$j^* = \max\{j \in [2, H] \mid \text{null}(\hat{W}_{j:2}) \supsetneq \text{null}(\hat{W}_{j-1:2})\}.$$

When you start from $j = H$ down to $j = 2$ and compare $\text{null}(\hat{W}_{j:2})$ and $\text{null}(\hat{W}_{j-1:2})$, the first iterate $j$ at which you have $\text{null}(\hat{W}_{j:2}) \neq \text{null}(\hat{W}_{j-1:2})$ is $j^*$. If all null spaces of matrices from $\hat{W}_{H:2}$ to $\hat{W}_2$ are equal, $j^* = 2$ which follows from the notational convention that $\text{null}(\hat{W}_{1:2}) = \text{null}(I_{d_1}) = \{0\}$. According to $j^*$, in Case 2 we perturb $\hat{W}_1, \hat{W}_{H+1}, \hat{W}_H, \ldots, \hat{W}_{j^*}$ to get $(P_j)_{j=1}^{H+1}$ and $(Q_j)_{j=1}^{H+1}$.

Recall the definition of left-null space of matrix $A$: $\text{leftnull}(A) = \{v \mid v^T A = 0\}$. By definition of $j^*$, note that

$$\text{null}(\hat{W}_{H+1:2}) = \text{null}(\hat{W}_{H:2}) = \cdots = \text{null}(\hat{W}_{j^*:2})$$
$$\Leftrightarrow \text{row}(\hat{W}_{H+1:2}) = \text{row}(\hat{W}_{H:2}) = \cdots = \text{row}(\hat{W}_{j^*:2})$$

$$\Leftrightarrow \operatorname{rank}(\hat{W}_{H+1:2}) = \operatorname{rank}(\hat{W}_{H:2}) = \cdots = \operatorname{rank}(\hat{W}_{j^*:2}),$$

which means the products are all rank-deficient (recall $\operatorname{rank}(\hat{W}_{H+1:2}) < d_y$ and all $d_j \geq d_y$), and hence they all have nontrivial left-null spaces $\operatorname{leftnull}(\hat{W}_{H:2}), \ldots, \operatorname{leftnull}(\hat{W}_{j^*:2})$ as well.

We choose some unit vectors as the following:

$$v_0 = [U_r]_{.,1},$$
$$v_1 \in \operatorname{null}(\hat{W}_{j^*:2}) \cap \operatorname{null}(\hat{W}_{j^*-1:2})^\perp,$$
$$v_{H+1} = [U_l]_{.,1},$$
$$v_H \in \operatorname{leftnull}(\hat{W}_{H:2}),$$
$$\cdots$$
$$v_{j^*} \in \operatorname{leftnull}(\hat{W}_{j^*:2}).$$

Then, for a $\gamma \in (0, \epsilon]$ whose value will be specified later, define

$$\Delta_1 := \gamma v_1 v_0^T \in \mathbb{R}^{d_1 \times d_x},$$
$$\Delta_{H+1} := \gamma v_{H+1} v_H^T \in \mathbb{R}^{d_y \times d_H},$$
$$\cdots$$
$$\Delta_{j^*+1} := \gamma v_{j^*+1} v_{j^*}^T \in \mathbb{R}^{d_{j^*+1} \times d_{j^*}},$$

and $V_j := \hat{W}_j + \Delta_j$ accordingly for $j = 1, j^*+1, \ldots, H+1$.

By definition of $\Delta_j$'s, note that

$$V_{H+1:j^*+1} \hat{W}_{j^*:2} V_1$$
$$= V_{H+1:j^*+2} \hat{W}_{j^*+1:2} V_1 + V_{H+1:j^*+2} \Delta_{j^*+1} \hat{W}_{j^*:2} V_1 = V_{H+1:j^*+2} \hat{W}_{j^*+1:2} V_1 \qquad (A.19)$$
$$= V_{H+1:j^*+3} \hat{W}_{j^*+2:2} V_1 + V_{H+1:j^*+3} \Delta_{j^*+2} \hat{W}_{j^*+1:2} V_1 = V_{H+1:j^*+3} \hat{W}_{j^*+2:2} V_1 \qquad (A.20)$$
$$= \cdots$$
$$= \hat{W}_{H+1:2} V_1 + \Delta_{H+1} \hat{W}_{H:2} V_1 = \hat{W}_{H+1:2} V_1 \qquad (A.21)$$
$$= \hat{W}_{H+1:1} + \hat{W}_{H+1:2} \Delta_1 = \hat{W}_{H+1:1}, \qquad (A.22)$$

where in (A.19) we used the definition that $v_{j^*} \in \operatorname{leftnull}(\hat{W}_{j^*:2})$, in (A.20) that $v_{j^*+1} \in \operatorname{leftnull}(\hat{W}_{j^*+1:2})$, in (A.21) that $v_H \in \operatorname{leftnull}(\hat{W}_{H:2})$, and in (A.22) that $v_1 \in \operatorname{null}(\hat{W}_{j^*:2})$.

Now consider the following matrix product:

$$(V_{H+1:j^*+1})^T \nabla \ell_0(\hat{W}_{H+1:1}) V_1^T (\hat{W}_{j^*-1:2})^T$$
$$= (\hat{W}_{j^*+1} + \Delta_{j^*+1})^T \cdots (\hat{W}_{H+1} + \Delta_{H+1})^T \nabla \ell_0(\hat{W}_{H+1:1})(\hat{W}_1 + \Delta_1)^T \hat{W}_2^T \cdots \hat{W}_{j^*-1}^T. \qquad (A.23)$$

We are going to show that for small enough $\gamma \in (0, \epsilon]$, this product is nonzero. If we expand (A.23), there are many terms in the summation. However, note that the expansion can be arranged in the following form:

$$(\hat{W}_{j^*+1} + \Delta_{j^*+1})^T \cdots (\hat{W}_{H+1} + \Delta_{H+1})^T \nabla \ell_0(\hat{W}_{H+1:1})(\hat{W}_1 + \Delta_1)^T \hat{W}_2^T \cdots \hat{W}_{j^*-1}^T$$
$$= C_0 + C_1 \gamma + C_2 \gamma^2 + \cdots + C_{H-j^*+2} \gamma^{H-j^*+2} \qquad (A.24)$$

where $C_j \in \mathbb{R}^{d_{j^*} \times d_{j^*-1}}$ for all $j$ and $C_j$ doesn't depend on $\gamma$, and specifically

$$C_0 = \hat{W}_{j^*+1}^T \cdots \hat{W}_{H+1}^T \nabla \ell_0(\hat{W}_{H+1:1}) \hat{W}_1^T \hat{W}_2^T \cdots \hat{W}_{j^*-1}^T,$$
$$C_{H-j^*+2} = \frac{1}{\gamma^{H-j^*+2}} \Delta_{j^*+1}^T \cdots \Delta_{H+1}^T \nabla \ell_0(\hat{W}_{H+1:1}) \Delta_1^T \hat{W}_2^T \cdots \hat{W}_{j^*-1}^T.$$

Because the $C_0$ is exactly equal to $\frac{\partial \ell}{\partial W_{j^*}}$ evaluated at a critical point $((\hat{W}_j)_{j=1}^{H+1})$, $C_0 = 0$. Also, due to definitions of $\Delta_j$'s,

$$C_{H-j^*+2} = (v_{j^*} v_{j^*+1}^T)(v_{j^*+1} v_{j^*+2}^T) \cdots (v_H v_{H+1}^T) \nabla \ell_0(\hat{W}_{H+1:1})(v_0 v_1^T)(\hat{W}_{j^*-1:2})^T$$

$$= v_{j^*} v_{H+1}^T \nabla \ell_0(\hat{W}_{H+1:1}) v_0 v_1^T (\hat{W}_{j^*-1:2})^T.$$

First, $v_{j^*}$ is a nonzero column vector. Since $v_{H+1} = [U_l]_{\cdot,1}$ and $v_0 = [U_r]_{\cdot,1}$, $v_{H+1}^T \nabla \ell_0(\hat{W}_{H+1:1}) v_0 = \sigma_{\max}(\nabla \ell_0(\hat{W}_{H+1:1})) > 0$. Also, since $v_1 \in \text{row}(\hat{W}_{j^*-1:2})$, $v_1^T (\hat{W}_{j^*-1:2})^T$ will be a nonzero row vector. Thus, the product $C_{H-j^*+2}$ will be nonzero.

Since $C_{H-j^*+2} \neq 0$, we can pick any index $(\alpha, \beta)$ such that the $(\alpha, \beta)$-th entry of $C_{H-j^*+2}$, denoted as $[C_{H-j^*+2}]_{\alpha,\beta}$, is nonzero. Then, the $(\alpha, \beta)$-th entry of (A.24) can be written as

$$c_1 \gamma + c_2 \gamma^2 + \cdots + c_{H-j^*+2} \gamma^{H-j^*+2}, \tag{A.25}$$

where $c_j = [C_j]_{\alpha,\beta}$. To show that the matrix product (A.23) is nonzero, it suffices to show that its $(\alpha, \beta)$-th entry (A.25) is nonzero. If $c_1 = \cdots = c_{H-j^*+1} = 0$, then with the choice of $\gamma = \epsilon$, (A.25) is trivially nonzero. If some of $c_1, \ldots, c_{H-j^*+1}$ are nonzero, we can scale $\gamma \in (0, \epsilon]$ arbitrarily small, so that

$$|c_1 \gamma + \cdots + c_{H-j^*+1} \gamma^{H-j^*+1}| > |c_{H-j^*+2} \gamma^{H-j^*+2}|,$$

and thus (A.25) can never be zero. From this, with sufficiently small $\gamma$, the matrix product (A.23) is nonzero.

Now define the perturbation on $\hat{W}_{j^*}$:

$$\Delta_{j^*} := \frac{\epsilon (V_{H+1:j^*+1})^T \nabla \ell_0(\hat{W}_{H+1:1}) V_1^T (\hat{W}_{j^*-1:2})^T}{\|(V_{H+1:j^*+1})^T \nabla \ell_0(\hat{W}_{H+1:1}) V_1^T (\hat{W}_{j^*-1:2})^T\|_\text{F}},$$

so that $\hat{W}_{j^*} + \Delta_{j^*} \in B_\epsilon(\hat{W}_{j^*})$. Then, observe that

$$\langle V_{H+1:j^*+1} \Delta_{j^*} \hat{W}_{j^*-1:2} V_1, \nabla \ell_0(\hat{W}_{H+1:1}) \rangle = \text{tr}((V_{H+1:j^*+1} \Delta_{j^*} \hat{W}_{j^*-1:2} V_1)^T \nabla \ell_0(\hat{W}_{H+1:1}))$$
$$= \text{tr}(\Delta_{j^*}^T (V_{H+1:j^*+1})^T \nabla \ell_0(\hat{W}_{H+1:1}) V_1^T (\hat{W}_{j^*-1:2})^T) = \langle \Delta_{j^*}, (V_{H+1:j^*+1})^T \nabla \ell_0(\hat{W}_{H+1:1}) V_1^T (\hat{W}_{j^*-1:2})^T \rangle > 0.$$

This means that $V_{H+1:j^*+1} \Delta_{j^*} \hat{W}_{j^*-1:2} V_1$ and $-V_{H+1:j^*+1} \Delta_{j^*} \hat{W}_{j^*-1:2} V_1$ are ascent and descent directions, respectively, of $\ell_0(R)$ at $\hat{W}_{H+1:1}$. After that, the proof is very similar to the previous case. We can define

$$(P_j)_{j=1}^{H+1} = (V_1, \hat{W}_2, \ldots, \hat{W}_{j^*-1}, \hat{W}_{j^*} + \eta \Delta_{j^*}, V_{j^*+1}, \ldots, V_{H+1}) \in \prod_{j=1}^{H+1} \mathcal{B}_\epsilon(\hat{W}_j)$$

$$(Q_j)_{j=1}^{H+1} = (V_1, \hat{W}_2, \ldots, \hat{W}_{j^*-1}, \hat{W}_{j^*} - \eta \Delta_{j^*}, V_{j^*+1}, \ldots, V_{H+1}) \in \prod_{j=1}^{H+1} \mathcal{B}_\epsilon(\hat{W}_j),$$

where $0 < \eta \leq 1$ is small enough, to show that by differentiability of $\ell_0(R)$, we get $\ell((P_j)_{j=1}^{H+1}) > \ell((\hat{W}_j)_{j=1}^{H+1}) > \ell((Q_j)_{j=1}^{H+1})$.

### A7.2 PROOF OF PART 1, IF $d_y \geq d_x$

First, note that $\nabla \ell_0(\hat{W}_{H+1:1})(\hat{W}_{H:1})^T = 0$, because it is $\frac{\partial \ell}{\partial W_{H+1}}$ evaluated at a critical point $(\hat{W}_j)_{j=1}^{H+1}$. This equation implies $\text{row}(\nabla \ell_0(\hat{W}_{H+1:1}))^\perp \supseteq \text{row}(\hat{W}_{H:1})$. Since $\nabla \ell_0(\hat{W}_{H+1:1}) \neq 0$, $\text{row}(\nabla \ell_0(\hat{W}_{H+1:1}))^\perp$ cannot be the whole $\mathbb{R}^{d_x}$, and it is a strict subspace of $\mathbb{R}^{d_x}$. Observe that $\hat{W}_{H:1} \in \mathbb{R}^{d_H \times d_x}$ and $d_x \leq d_H$. Since $\text{row}(\hat{W}_{H:1}) \subseteq \text{row}(\nabla \ell_0(\hat{W}_{H+1:1}))^\perp \subsetneq \mathbb{R}^{d_x}$, this means $\text{rank}(\hat{W}_{H:1}) < d_x$, hence $\text{leftnull}(\hat{W}_{H:1})$ is not a trivial subspace.

Now observe that

$$\text{leftnull}(\hat{W}_{H:1}) \supseteq \text{leftnull}(\hat{W}_{H:2}) \supseteq \cdots \supseteq \text{leftnull}(\hat{W}_H),$$

where some of left-null spaces in the right could be zero-dimensional. The procedure of choosing the perturbation depends on these left-null spaces. We can split the proof into two cases: $\text{leftnull}(\hat{W}_{H:1}) \neq \text{leftnull}(\hat{W}_{H:2})$ and $\text{leftnull}(\hat{W}_{H:1}) = \text{leftnull}(\hat{W}_{H:2})$. Because the former case is simpler, we prove the former case first.

Before we dive in, again take SVD of $\nabla \ell_0(\hat{W}_{H+1:1}) = U_l \Sigma U_r^T$. Since $\nabla \ell_0(\hat{W}_{H+1:1}) \neq 0$, there is at least one positive singular value, so $\sigma_{\max}(\nabla \ell_0(\hat{W}_{H+1:1})) > 0$. Recall the notation that $[U_l]_{\cdot,1}$ and $[U_r]_{\cdot,1}$ are first column vectors of $U_l$ and $U_r$, respectively.

**Case 1:** $\mathrm{leftnull}(\hat{W}_{H:1}) \neq \mathrm{leftnull}(\hat{W}_{H:2})$. In this case, $\mathrm{leftnull}(\hat{W}_{H:1}) \supsetneq \mathrm{leftnull}(\hat{W}_{H:2})$. We will perturb $\hat{W}_1$ and $\hat{W}_{H+1}$ to obtain the desired tuples $(P_j)_{j=1}^{H+1}$ and $(Q_j)_{j=1}^{H+1}$.

Now choose two unit vectors $v_H$ and $v_{H+1}$, as the following:

$$v_H \in \mathrm{leftnull}(\hat{W}_{H:1}) \cap \mathrm{leftnull}(\hat{W}_{H:2})^{\perp}, \ v_{H+1} = [U_l]_{\cdot,1},$$

and then define $\Delta_{H+1} := \epsilon v_{H+1} v_H^T \in \mathbb{R}^{d_y \times d_H}$, and $V_{H+1} := \hat{W}_{H+1} + \Delta_{H+1}$. We can check $V_{H+1} \in \mathcal{B}_{\epsilon}(\hat{W}_{H+1})$ from the fact that $v_H$ and $v_{H+1}$ are unit vectors. Since $v_H \in \mathrm{leftnull}(\hat{W}_{H:1})$, observe that

$$V_{H+1}\hat{W}_{H:1} = \hat{W}_{H+1:1} + \epsilon v_{H+1} v_H^T \hat{W}_{H:1} = \hat{W}_{H+1:1}.$$

With this definition of $V_{H+1}$, we can also see that

$$(\hat{W}_{H:2})^T V_{H+1}^T \nabla \ell_0(\hat{W}_{H+1:1}) = (\hat{W}_{H+1:2})^T \nabla \ell_0(\hat{W}_{H+1:1}) + \epsilon(\hat{W}_{H:2})^T v_H v_{H+1}^T \nabla \ell_0(\hat{W}_{H+1:1}).$$

Note that $(\hat{W}_{H+1:2})^T \nabla \ell_0(\hat{W}_{H+1:1})$ is exactly equal to $\frac{\partial \ell}{\partial W_1}$ evaluated at $(\hat{W}_j)_{j=1}^{H+1}$, hence is zero by assumption that $(\hat{W}_j)_{j=1}^{H+1}$ is a critical point. Since $v_H \in \mathrm{leftnull}(\hat{W}_{H:2})^{\perp} = \mathrm{col}(\hat{W}_{H:2})$, $(\hat{W}_{H:2})^T v_H$ is a nonzero column vector, and since $v_{H+1} = [U_l]_{\cdot,1}$, $v_{H+1}^T \nabla \ell_0(\hat{W}_{H+1:1}) = \sigma_{\max}(\nabla \ell_0(\hat{W}_{H+1:1}))([U_r]_{\cdot,1})^T$, which is a nonzero row vector. From this observation, we can see that $(\hat{W}_{H:2})^T v_H v_{H+1}^T \nabla \ell_0(\hat{W}_{H+1:1})$ is nonzero, and so is $(\hat{W}_{H:2})^T V_{H+1}^T \nabla \ell_0(\hat{W}_{H+1:1})$.

Now define the perturbation on $\hat{W}_1$:

$$\Delta_1 := \frac{\epsilon(\hat{W}_{H:2})^T V_{H+1}^T \nabla \ell_0(\hat{W}_{H+1:1})}{\|(\hat{W}_{H:2})^T V_{H+1}^T \nabla \ell_0(\hat{W}_{H+1:1})\|_{\mathrm{F}}},$$

so that $\hat{W}_1 + \Delta_1 \in B_{\epsilon}(\hat{W}_1)$. Then, observe that

$$\langle V_{H+1}\hat{W}_{H:2}\Delta_1, \nabla \ell_0(\hat{W}_{H+1:1}) \rangle = \mathrm{tr}((V_{H+1}\hat{W}_{H:2}\Delta_1)^T \nabla \ell_0(\hat{W}_{H+1:1}))$$
$$= \mathrm{tr}(\Delta_1^T (\hat{W}_{H:2})^T V_{H+1}^T \nabla \ell_0(\hat{W}_{H+1:1})) = \langle \Delta_1, (\hat{W}_{H:2})^T V_{H+1}^T \nabla \ell_0(\hat{W}_{H+1:1}) \rangle > 0,$$

by definition of $\Delta_1$. This means that $V_{H+1}\hat{W}_{H:2}\Delta_1$ and $-V_{H+1}\hat{W}_{H:2}\Delta_1$ are ascent and descent directions, respectively, of $\ell_0(R)$ at $\hat{W}_{H+1:1}$. Since $\ell_0$ is a differentiable function, there exists small enough $0 < \eta \leq 1$ that satisfies

$$\ell_0(\hat{W}_{H+1:1} + \eta V_{H+1}\hat{W}_{H:2}\Delta_1) > \ell_0(\hat{W}_{H+1:1}),$$
$$\ell_0(\hat{W}_{H+1:1} - \eta V_{H+1}\hat{W}_{H:2}\Delta_1) < \ell_0(\hat{W}_{H+1:1}).$$

Now define

$$(P_j)_{j=1}^{H+1} = (\hat{W}_1 + \eta\Delta_1, \hat{W}_2, \dots, \hat{W}_H, V_{H+1}),$$
$$(Q_j)_{j=1}^{H+1} = (\hat{W}_1 - \eta\Delta_1, \hat{W}_2, \dots, \hat{W}_H, V_{H+1}).$$

We can check $(P_j)_{j=1}^{H+1}, (Q_j)_{j=1}^{H+1} \in \prod_{j=1}^{H+1} \mathcal{B}_{\epsilon}(\hat{W}_j)$, and

$$P_{H+1:1} = \hat{W}_{H+1:1} + \eta V_{H+1}\hat{W}_{H:2}\Delta_1.$$
$$Q_{H+1:1} = \hat{W}_{H+1:1} - \eta V_{H+1}\hat{W}_{H:2}\Delta_1.$$

By definition of $\ell((W_j)_{j=1}^{H+1})$, this shows that $\ell((P_j)_{j=1}^{H+1}) > \ell((\hat{W}_j)_{j=1}^{H+1}) > \ell((Q_j)_{j=1}^{H+1})$. This construction holds for any $\epsilon > 0$, proving that $(\hat{W}_j)_{j=1}^{H+1}$ can be neither a local maximum nor a local minimum.

**Case 2:** $\mathrm{leftnull}(\hat{W}_{H:1}) = \mathrm{leftnull}(\hat{W}_{H:2})$. By and large, the proof of this case goes the same, except that we need a little more care on what perturbations to make. Define

$$j^* = \min\{j \in [2, H] \mid \mathrm{leftnull}(\hat{W}_{H:j}) \supsetneq \mathrm{leftnull}(\hat{W}_{H:j+1})\}.$$

When you start from $j = 2$ up to $j = H$ and compare $\mathrm{leftnull}(\hat{W}_{H:j})$ and $\mathrm{leftnull}(\hat{W}_{H:j+1})$, the first iterate $j$ at which you have $\mathrm{leftnull}(\hat{W}_{H:j}) \neq \mathrm{leftnull}(\hat{W}_{H:j+1})$ is $j^*$. If all left-null spaces of matrices from $\hat{W}_{H:2}$ to $\hat{W}_H$ are equal, $j^* = H$ which follows from the notational convention that $\mathrm{leftnull}(\hat{W}_{H:H+1}) = \mathrm{leftnull}(I_{d_H}) = \{0\}$. According to $j^*$, in Case 2 we perturb $\hat{W}_{H+1}, \hat{W}_1, \hat{W}_2,$ $\ldots, \hat{W}_{j^*}$ to get $(P_j)_{j=1}^{H+1}$ and $(Q_j)_{j=1}^{H+1}$.

By definition of $j^*$, note that

$$\mathrm{leftnull}(\hat{W}_{H:1}) = \mathrm{leftnull}(\hat{W}_{H:2}) = \cdots = \mathrm{leftnull}(\hat{W}_{H:j^*})$$
$$\Leftrightarrow \mathrm{col}(\hat{W}_{H:1}) = \mathrm{col}(\hat{W}_{H:2}) = \cdots = \mathrm{col}(\hat{W}_{H:j^*})$$
$$\Leftrightarrow \mathrm{rank}(\hat{W}_{H:1}) = \mathrm{rank}(\hat{W}_{H:2}) = \cdots = \mathrm{rank}(\hat{W}_{H:j^*})$$

which means the products are all rank-deficient (recall $\mathrm{rank}(\hat{W}_{H:1}) < d_x$ and all $d_j \geq d_x$), and hence they all have nontrivial null spaces $\mathrm{null}(\hat{W}_{H:2}), \ldots, \mathrm{null}(\hat{W}_{H:j^*})$ as well.

We choose some unit vectors as the following:

$$v_0 = [U_r]_{.,1},$$
$$v_1 \in \mathrm{null}(\hat{W}_{H:2}),$$
$$\cdots$$
$$v_{j^*-1} \in \mathrm{null}(\hat{W}_{H:j^*})$$
$$v_H \in \mathrm{leftnull}(\hat{W}_{H:j^*}) \cap \mathrm{leftnull}(\hat{W}_{H:j^*+1})^{\perp},$$
$$v_{H+1} = [U_l]_{.,1}.$$

Then, for a $\gamma \in (0, \epsilon]$ whose value will be specified later, define

$$\Delta_1 := \gamma v_1 v_0^T \in \mathbb{R}^{d_1 \times d_x},$$
$$\cdots$$
$$\Delta_{j^*-1} := \gamma v_{j^*-1} v_{j^*-2}^T \in \mathbb{R}^{d_{j^*-1} \times d_{j^*-2}},$$
$$\Delta_{H+1} := \gamma v_{H+1} v_H^T \in \mathbb{R}^{d_y \times d_H},$$

and $V_j := \hat{W}_j + \Delta_j$ accordingly for $j = 1, \ldots, j^* - 1, H + 1$.

By definition of $\Delta_j$'s, note that

$$V_{H+1} \hat{W}_{H:j^*} V_{j^*-1:1}$$
$$= V_{H+1} \hat{W}_{H:j^*-1} V_{j^*-2:1} + V_{H+1} \hat{W}_{H:j^*} \Delta_{j^*-1} V_{j^*-2:1} = V_{H+1} \hat{W}_{H:j^*-1} V_{j^*-2:1} \quad \text{(A.26)}$$
$$= V_{H+1} \hat{W}_{H:j^*-2} V_{j^*-3:1} + V_{H+1} \hat{W}_{H:j^*-1} \Delta_{j^*-2} V_{j^*-3:1} = V_{H+1} \hat{W}_{H:j^*-2} V_{j^*-3:1} \quad \text{(A.27)}$$
$$= \cdots$$
$$= V_{H+1} \hat{W}_{H:1} + V_{H+1} \hat{W}_{H:2} \Delta_1 = V_{H+1} \hat{W}_{H:1} \quad \text{(A.28)}$$
$$= \hat{W}_{H+1:1} + \Delta_{H+1} \hat{W}_{H:1} = \hat{W}_{H+1:1}, \quad \text{(A.29)}$$

where in (A.26) we used the definition that $v_{j^*-1} \in \mathrm{null}(\hat{W}_{H:j^*})$, in (A.27) that $v_{j^*-2} \in \mathrm{null}(\hat{W}_{H:j^*-1})$, in (A.28) that $v_1 \in \mathrm{null}(\hat{W}_{H:2})$, and in (A.29) that $v_H \in \mathrm{leftnull}(\hat{W}_{H:j^*})$.

Now consider the following matrix product:

$$(\hat{W}_{H:j^*+1})^T V_{H+1}^T \nabla \ell_0 (\hat{W}_{H+1:1}) (V_{j^*-1:1})^T$$
$$= (\hat{W}_{H:j^*+1})^T (\hat{W}_{H+1} + \Delta_{H+1})^T \nabla \ell_0 (\hat{W}_{H+1:1}) (\hat{W}_1 + \Delta_1)^T \cdots (\hat{W}_{j^*-1} + \Delta_{j^*-1})^T. \quad \text{(A.30)}$$

We are going to show that for small enough $\gamma \in (0, \epsilon]$, this product is nonzero. If we expand (A.30), there are many terms in the summation. However, note that the expansion can be arranged in the following form:

$$(\hat{W}_{H:j^*+1})^T (\hat{W}_{H+1} + \Delta_{H+1})^T \nabla \ell_0 (\hat{W}_{H+1:1}) (\hat{W}_1 + \Delta_1)^T \cdots (\hat{W}_{j^*-1} + \Delta_{j^*-1})^T$$

$$=C_0 + C_1\gamma + C_2\gamma^2 + \cdots + C_{j^*}\gamma^{j^*} \tag{A.31}$$

where $C_j \in \mathbb{R}^{d_{j^*} \times d_{j^*-1}}$ for all $j$ and $C_j$ doesn't depend on $\gamma$, and specifically

$$C_0 = \hat{W}_{j^*+1}^T \cdots \hat{W}_{H+1}^T \nabla \ell_0(\hat{W}_{H+1:1})\hat{W}_1^T \hat{W}_2^T \cdots \hat{W}_{j^*-1}^T,$$

$$C_{j^*} = \frac{1}{\gamma^{j^*}} \hat{W}_{j^*+1}^T \cdots \hat{W}_H^T \Delta_{H+1}^T \nabla \ell_0(\hat{W}_{H+1:1})\Delta_1^T \cdots \Delta_{j^*-1}^T.$$

Because the $C_0$ is exactly equal to $\frac{\partial \ell}{\partial W_{j^*}}$ evaluated at a critical point $((\hat{W}_j)_{j=1}^{H+1})$, $C_0 = 0$. Also, due to definitions of $\Delta_j$'s,

$$C_{j^*} = (\hat{W}_{H:j^*+1})^T (v_H v_{H+1}^T)\nabla \ell_0(\hat{W}_{H+1:1})(v_0 v_1^T)(v_1 v_2^T) \cdots (v_{j^*-2} v_{j^*-1}^T)$$

$$= (\hat{W}_{H:j^*+1})^T v_H v_{H+1}^T \nabla \ell_0(\hat{W}_{H+1:1})v_0 v_{j^*-1}^T.$$

First, since $v_H \in \text{col}(\hat{W}_{H:j^*+1})$, $(\hat{W}_{H:j^*+1})^T v_H$ is a nonzero column vector. Also, since $v_{H+1} = [U_l]_{\cdot,1}$ and $v_0 = [U_r]_{\cdot,1}$, the product $v_{H+1}^T \nabla \ell_0(\hat{W}_{H+1:1})v_0 = \sigma_{\max}(\nabla \ell_0(\hat{W}_{H+1:1})) > 0$. Finally, $v_{j^*-1}^T$ is a nonzero row vector. Thus, the product $C_{j^*}$ will be nonzero.

Since $C_{j^*} \neq 0$, we can pick any index $(\alpha, \beta)$ such that the $(\alpha, \beta)$-th entry of $C_{j^*}$, denoted as $[C_{j^*}]_{\alpha,\beta}$, is nonzero. Then, the $(\alpha, \beta)$-th entry of (A.31) can be written as

$$c_1\gamma + c_2\gamma^2 + \cdots + c_{j^*}\gamma^{j^*}, \tag{A.32}$$

where $c_j = [C_j]_{\alpha,\beta}$. To show that the matrix product (A.30) is nonzero, it suffices to show that its $(\alpha, \beta)$-th entry (A.32) is nonzero. If $c_1 = \cdots = c_{j^*-1} = 0$, then with the choice of $\gamma = \epsilon$, (A.32) is trivially nonzero. If some of $c_1, \ldots, c_{j^*-1}$ are nonzero, we can scale $\gamma \in (0, \epsilon]$ arbitrarily small, so that

$$|c_1\gamma + \cdots + c_{j^*-1}\gamma^{j^*-1}| > |c_{j^*}\gamma^{j^*}|,$$

and thus (A.32) can never be zero. From this, with sufficiently small $\gamma$, the matrix product (A.30) is nonzero.

Now define the perturbation on $\hat{W}_{j^*}$:

$$\Delta_{j^*} := \frac{\epsilon(\hat{W}_{H:j^*+1})^T V_{H+1}^T \nabla \ell_0(\hat{W}_{H+1:1})(V_{j^*-1:1})^T}{\|(\hat{W}_{H:j^*+1})^T V_{H+1}^T \nabla \ell_0(\hat{W}_{H+1:1})(V_{j^*-1:1})^T\|_{\mathrm{F}}},$$

so that $\hat{W}_{j^*} + \Delta_{j^*} \in B_\epsilon(\hat{W}_{j^*})$. Then, observe that

$$\langle V_{H+1}\hat{W}_{H:j^*+1}\Delta_{j^*}V_{j^*-1:1}, \nabla \ell_0(\hat{W}_{H+1:1})\rangle = \text{tr}((V_{H+1}\hat{W}_{H:j^*+1}\Delta_{j^*}V_{j^*-1:1})^T \nabla \ell_0(\hat{W}_{H+1:1}))$$

$$= \text{tr}(\Delta_{j^*}^T(\hat{W}_{H:j^*+1})^T V_{H+1}^T \nabla \ell_0(\hat{W}_{H+1:1})(V_{j^*-1:1})^T) = \langle \Delta_{j^*}, (\hat{W}_{H:j^*+1})^T V_{H+1}^T \nabla \ell_0(\hat{W}_{H+1:1})(V_{j^*-1:1})^T\rangle > 0.$$

This means that $V_{H+1}\hat{W}_{H:j^*+1}\Delta_{j^*}V_{j^*-1:1}$ and $-V_{H+1}\hat{W}_{H:j^*+1}\Delta_{j^*}V_{j^*-1:1}$ are ascent and descent directions, respectively, of $\ell_0(R)$ at $\hat{W}_{H+1:1}$. After that, the proof is very similar to the previous case. We can define

$$(P_j)_{j=1}^{H+1} = (V_1, \ldots, V_{j^*-1}, \hat{W}_{j^*} + \eta\Delta_{j^*}, \hat{W}_{j^*+1}, \ldots, \hat{W}_H, V_{H+1}) \in \prod_{j=1}^{H+1} \mathcal{B}_\epsilon(\hat{W}_j)$$

$$(Q_j)_{j=1}^{H+1} = (V_1, \ldots, V_{j^*-1}, \hat{W}_{j^*} - \eta\Delta_{j^*}, \hat{W}_{j^*+1}, \ldots, \hat{W}_H, V_{H+1}) \in \prod_{j=1}^{H+1} \mathcal{B}_\epsilon(\hat{W}_j),$$

where $0 < \eta \leq 1$ is small enough, to show that by differentiability of $\ell_0(R)$, we get $\ell((P_j)_{j=1}^{H+1}) > \ell((\hat{W}_j)_{j=1}^{H+1}) > \ell((Q_j)_{j=1}^{H+1})$.

## A7.3 PROOF OF PART 2(A)

In this part, we show that if $\nabla \ell_0(\hat{W}_{H+1:1}) = 0$ and $\hat{W}_{H+1:1}$ is a local min of $\ell_0$, then $(\hat{W}_j)_{j=1}^{H+1}$ is a local min of $\ell$. The proof for local max case can be done in a very similar way.

Since $\hat{W}_{H+1:1}$ is a local minimum of $\ell_0$, there exists $\epsilon > 0$ such that, for any $R$ satisfying $\|R - \hat{W}_{H+1:1}\|_F \le \epsilon$, we have $\ell_0(R) \ge \ell_0(\hat{W}_{H+1:1})$. We prove that $(\hat{W}_j)_{j=1}^{H+1}$ is a local minimum of $\ell$ by showing that there exists a neighborhood of $(\hat{W}_j)_{j=1}^{H+1}$ in which any point $(V_j)_{j=1}^{H+1}$ satisfies $\ell((V_j)_{j=1}^{H+1}) \ge \ell((\hat{W}_j)_{j=1}^{H+1})$.

Now define

$$0 < \epsilon_j \le \frac{\epsilon}{2(H+1)\max\left\{\|\hat{W}_{H+1:j+1}\|_F \|\hat{W}_{j-1:1}\|_F, 1\right\}}.$$

Observe that $\frac{a}{\max\{a,1\}} \le 1$ for $a \ge 0$. Then, for all $j \in [H+1]$, pick any $V_j$ such that $\|V_j - \hat{W}_j\|_F \le \epsilon_j$. Denote $\Delta_j = V_j - \hat{W}_j$ for all $j$. Now, by triangle inequality and submultiplicativity of Frobenius norm,

$$
\begin{aligned}
\|(\hat{W}_{H+1} + \Delta_{H+1})\cdots(\hat{W}_1 + \Delta_1) - \hat{W}_{H+1:1}\|_F &\le \sum_{j=1}^{H+1} \|\hat{W}_{H+1:j+1}\Delta_j \hat{W}_{j-1:1}\|_F + O(\max_j \|\Delta_j\|_F^2) \\
&\le \sum_{j=1}^{H+1} \|\hat{W}_{H+1:j+1}\|_F \|\Delta_j\|_F \|\hat{W}_{j-1:1}\|_F + O(\max_j \epsilon_j^2) \\
&\le \frac{\epsilon}{2} + O(\max_j \epsilon_j^2) \le \epsilon,
\end{aligned}
$$

for small enough $\epsilon_j$'s.

Given this, for any $(V_j)_{j=1}^{H+1}$ in the neighborhood of $(\hat{W}_j)_{j=1}^{H+1}$ defined by $\epsilon_j$'s, $\|V_{H+1:1} - \hat{W}_{H+1:1}\|_F \le \epsilon$, so $\ell_0(V_{H+1:1}) \ge \ell_0(\hat{W}_{H+1:1})$, meaning $\ell((V_j)_{j=1}^{H+1}) \ge \ell((\hat{W}_j)_{j=1}^{H+1})$. Thus, $(\hat{W}_j)_{j=1}^{H+1}$ is a local minimum of $\ell$.

### A7.4 PROOF OF PART 2(B)

For this part, we want to show that if $\nabla\ell_0(\hat{W}_{H+1:1}) = 0$, then $(\hat{W}_j)_{j=1}^{H+1}$ is a global min (or max) of $\ell$ if and only if $\hat{W}_{H+1:1}$ is a global min (or max) of $\ell_0$. We prove this by showing the following: if $d_j \ge \min\{d_x, d_y\}$ for all $j \in [H]$, for any $R \in \mathbb{R}^{d_y \times d_x}$ there exists a decomposition $(W_j)_{j=1}^{H+1}$ such that $R = W_{H+1:1}$.

We divide the proof into two cases: $d_x \ge d_y$ and $d_y \ge d_x$.

**Case 1: $d_x \ge d_y$.** If $d_x \ge d_y$, by assumption $d_j \ge d_y$ for all $j \in [H]$. Recall that $W_1 \in \mathbb{R}^{d_1 \times d_x}$. Given $R \in \mathbb{R}^{d_y \times d_x}$, we can fill the first $d_y$ rows of $W_1$ with $R$ and let any other entries be zero. For all the other matrices $W_2, \ldots, W_{H+1}$, we put ones to the diagonal entries while putting zeros to all the other entries. We can check that, by this construction, $R = W_{H+1:1}$ for this given $R$.

**Case 2: $d_y \ge d_x$.** If $d_y \ge d_x$, we have $d_j \ge d_x$ for all $j \in [H]$. Recall $W_{H+1} \in \mathbb{R}^{d_y \times d_H}$. Given $R \in \mathbb{R}^{d_y \times d_x}$, we can fill the first $d_x$ columns of $W_{H+1}$ with $R$ and let any other entries be zero. For all the other matrices $W_1, \ldots, W_H$, we put ones to the diagonal entries while putting zeros to all the other entries. By this construction, $R = W_{H+1:1}$ for given $R$.

Once this fact is given, by $\ell((W_j)_{j=1}^{H+1}) = \ell_0(W_{H+1:1})$,

$$\inf_R \ell_0(R) = \inf_{W_{H+1:1}} \ell_0(W_{H+1:1}) = \inf_{(W_j)_{j=1}^{H+1}} \ell((W_j)_{j=1}^{H+1}).$$

Thus, any $(\hat{W}_j)_{j=1}^{H+1}$ attaining a global min of $\ell$ must have $\inf_R \ell_0(R) = \ell_0(\hat{W}_{H+1:1})$, so $\hat{W}_{H+1:1}$ is also a global min of $\ell_0(R)$. Conversely, if $\ell_0(\hat{W}_{H+1:1}) = \inf \ell_0(R)$, then $\ell((\hat{W}_j)_{j=1}^{H+1}) = \inf \ell((W_j)_{j=1}^{H+1})$, so $(\hat{W}_j)_{j=1}^{H+1}$ is a global min of $\ell$. We can prove the global max case similarly.

A7.5   PROOF OF PART 3 AND 3(A)

Suppose there exists $j^* \in [H+1]$ such that $\hat{W}_{H+1:j^*+1}$ has full row rank and $\hat{W}_{j^*-1:1}$ has full column rank. For simplicity, define $A := \hat{W}_{H+1:j^*+1}$ and $B := \hat{W}_{j^*-1:1}$. Since $A^T$ has linearly independent columns, $B^T$ has linearly independent rows, and $\partial \ell / \partial W_{j^*} = 0$ at $(\hat{W}_j)_{j=1}^{H+1}$, $A^T \nabla \ell_0(\hat{W}_{H+1:1}) B^T = 0 \implies \nabla \ell_0(\hat{W}_{H+1:1}) = 0$, hence Parts 2(a) and 2(b) are implied.

For Part 3(a), we want to prove that if $(\hat{W}_j)_{j=1}^{H+1}$ is a local min of $\ell$, then $\hat{W}_{H+1:1}$ is a local min of $\ell_0$. By definition of local min, $\exists \epsilon > 0$ such that, for any $(V_j)_{j=1}^{H+1}$ for which $\|V_j - \hat{W}_j\|_F \le \epsilon$ (for $j \in [H+1]$), we have $\ell((V_j)_{j=1}^{H+1}) \ge \ell((\hat{W}_j)_{j=1}^{H+1})$. To show that $\hat{W}_{H+1:1}$ is a local min of $\ell_0$, we have to show there exists a neighborhood of $\hat{W}_{H+1:1}$ such that, any point $R$ in that neighborhood satisfies $\ell_0(R) \ge \ell_0(\hat{W}_{H+1:1})$. To prove this, we state the following lemma:

**Lemma A.6.** *Suppose $A := \hat{W}_{H+1:j^*+1}$ has full row rank and $B := \hat{W}_{j^*-1:1}$ has full column rank. Then, any $R$ satisfying $\|R - \hat{W}_{H+1:1}\|_F \le \sigma_{\min}(A)\sigma_{\min}(B)\epsilon$ can be decomposed into $R = V_{H+1:1}$, where*

$$V_{j^*} = \hat{W}_{j^*} + A^T(AA^T)^{-1}(R - \hat{W}_{H+1:1})(B^T B)^{-1} B^T,$$

*and $V_j = \hat{W}_j$ for $j \ne j^*$. Also, $\|V_j - \hat{W}_j\|_F \le \epsilon$ for all $j$.*

**Proof**   Since $A := \hat{W}_{H+1:j^*+1}$ has full row rank and $B := \hat{W}_{j^*-1:1}$ has full column rank, $\sigma_{\min}(A) > 0$, $\sigma_{\min}(B) > 0$, and $AA^T$ and $B^T B$ are invertible. Consider any $R$ satisfying $\|R - \hat{W}_{H+1:1}\|_F \le \sigma_{\min}(A)\sigma_{\min}(B)\epsilon$. Given the definitions of $V_j$'s in the statement of the lemma, we can check the identity that $R = V_{H+1:1}$ by

$$V_{H+1:1} = AV_j B = A\hat{W}_j B + (R - \hat{W}_{H+1:1}) = \hat{W}_{H+1:1} + (R - \hat{W}_{H+1:1}) = R.$$

Now It is left to show that $\|V_{j^*} - \hat{W}_{j^*}\|_F \le \epsilon$, so that $(V_j)_{j=1}^{H+1}$ indeed satisfies $\|V_j - \hat{W}_j\|_F \le \epsilon$ for all $j$. We can show that

$$\sigma_{\max}(A^T(AA^T)^{-1}) = 1/\sigma_{\min}(A), \ \sigma_{\max}((B^T B)^{-1}B^T) = 1/\sigma_{\min}(B).$$

Therefore,

$$
\begin{aligned}
\|V_{j^*} - \hat{W}_{j^*}\|_F &= \|A^T(AA^T)^{-1}(R - \hat{W}_{H+1:1})(B^T B)^{-1}B^T\|_F \\
&\le \sigma_{\max}(A^T(AA^T)^{-1})\sigma_{\max}((B^T B)^{-1}B^T)\|R - \hat{W}_{H+1:1}\|_F \\
&\le \frac{1}{\sigma_{\min}(A)\sigma_{\min}(B)} \cdot \sigma_{\min}(A)\sigma_{\min}(B)\epsilon = \epsilon.
\end{aligned}
$$

$\square$

The lemma shows that for any $R = V_{H+1:1}$ satisfying $\|R - \hat{W}_{H+1:1}\|_F \le \sigma_{\min}(A)\sigma_{\min}(B)\epsilon$, we have $\ell_0(R) = \ell_0(V_{H+1:1}) = \ell((V_j)_{j=1}^{H+1}) \ge \ell((\hat{W}_j)_{j=1}^{H+1}) = \ell_0(\hat{W}_{H+1:1})$. We can prove the local maximum part by a similar argument.

