# OpenReview forum: "Small nonlinearities in activation functions create bad local minima in neural networks"
_ICLR.cc/2019/Conference_

### Official Review · AnonReviewer1 · 2018-10-30
**elegant construction; interesting phenomenon**

**Rating:** 8
**Confidence:** 4

**Review:**


The authors provide a clean and easily understood sufficient
condition for spurious local minima to exist in networks with
a hidden layer using ReLUs or leaky ReLUs.  This condition,
that there is not linear transformation with zero loss,
is satisfied for almost all inputs with more examples than
input variables.

The construction is elegant.  The mathematical writing in the paper,
especially describing the proof of Theorem 1, is very nice -- they
expose the main ideas effectively.

I do not know of another paper using a similar proof, but I have not
studied the proofs of the most closely related papers prior to doing
this review, so I have limited ability to vouch for this paper's
technical novelty.

The authors also show that networks using many other popular
activation functions have spurious local minima for a very
simple dataset.  All of these analysis are unified using a
simple, if technical, set of conditions on activation function.

Finally, the authors prove a somewhat technical theorem about
optima in deep linear networks, which generalizes some
earlier treatments of this topic, providing an checkable
condition for global minimality.

There is extensive discussion of related work.  I am not aware of
related work not covered by the authors.

In some cases, when the authors discuss previous work, they write as
if restriction to the realizable case is an assumption, when it seems
to me to be more of a constraint.  In other words, it seems harder to
prove the existence of spurious minima in the realizable case.
They seem to acknowledge this after their statement of their Theorem 2,
which also uses a realizable dataset.

Also, a few papers, including the Venturi, et al paper cited by
the authors, have analyzed whether spurious local minima exist
in subsets of the parameter space, including those likely to
be reached during training with different sorts of initializations.
In light of this work, the authors might want to tone down claims
about how their work shows that results about linear networks do
not generalize to the non-linear case.  In particular, to make
their construction work in the case of wide networks, they
need an overwhelming majority of the hidden units to be "dead",
which seems as it is unlikely to arise from training with
commonly used initializations.

Overall, I think that this paper makes an interesting and
non-obvious contribution on a hot topic.

---

> ### Author Response · Authors · 2018-11-22
> **Reply to AnonReviewer1**
>
> We thank the reviewer for the enlightening feedback, and we are glad that the reviewer found our results/proof techniques interesting.
>
> We will address the comments/concerns below:
>
> * Regarding realizable datasets:
> We agree with the reviewer that proving existence of spurious local minima for realizable datasets is more difficult than non-realizable ones. When we discuss previous works, what we want to emphasize is that our Theorem 1 holds for *more general* datasets than existing results, without posing assumptions such as realizability. Our Theorem 1 holds for a set of datasets that do not fit linear models, which includes all non-realizable datasets and also *many* realizable datasets. To see why, note that perfect fit to linear models implies realizability, due to a construction similar to our Section 2.2, step 1.
>
> * Regarding other related results:
> First, we would like to emphasize that our construction of bad local minima doesn’t require the hidden units to be zero at the same time. They were set to zero just to simplify exposition. As noted at the end of Section 2.1, as long as the bias vector of first hidden node is a large positive number so that $W_1 X + b_1 1_m^T > 0$ entry-wise, the network behaves as a “locally” linear neural network, so any local minima in such a region can only do as well as a linear model, being a bad local minimum. Note that for any one-hidden-layer neural network (no matter how wide), there is a nontrivial (measure > 0) set of such points in the parameter space. We will make this point more explicitly as we revise the paper.
> Our focus in this paper is the existence/nonexistence of spurious local minima in the whole parameter space. We fully agree that the optimization algorithms with proper initialization may not reach such “bad” regions, and that when we restrict the parameter space to some subsets, things may be different. As suggested by the reviewer, we will adjust the claims in our next version of the paper.

---

### Official Review · AnonReviewer3 · 2018-11-05
**Review of "Small nonlinearities in activation functions create bad local minima in neural networks"**

**Rating:** 7
**Confidence:** 3

**Review:**

Paper represents theoretical analysis of the loss surface of neural networks. The authors supplied interesting results about local minima properties of neural networks. The paper is written quite well and easy to follow. Furthermore, authors made a comprehensive literature survey and connected their paper to already existing literature. The proofs seem correct (please note that paper is quite long and it requires more time then conference review period, hence, I stated “seem correct”).
Although, paper provides novel theorems, I have several concerns, and these are:
•	Isn’t it clear that (generally speaking) a non-convex problem will have many local minima? Previous paper in neural network community is (in my opinion) not theorems. I believe one should read those statements such that in practice (please note here practice means that architectures, e.g. resnet50, inceptionv3, used in day to day life) researchers don’t not observe those “really bad” local minima.
•	In ML literature, we have convex machines which are guaranteed to converge to global optimum. Given image dataset, or text datasets supervised deep learning in in general much better than convex methods e.g. SVMs. The paper clearly shows that there exist, in some cases, exponentially many local minima however  current training methods are able to find better solutions than convex methods (I am completely aware that functions classes are different however success metric, accuracy, is the same). Hence, how relevant are the results without taking in to account architectural choices or optimization methods for deep learning? May be structural risk minimisation is a better approach than empirical risk minimization for quantifying the performance of deep neural networks,
In conclusion, paper is interesting however I believe it need to be improved.

---

> ### Author Response · Authors · 2018-11-22
> **Reply to AnonReviewer3**
>
> We appreciate the reviewer’s time and thoughtful comments. We are glad that you found our paper easy to read. Below, we will provide answers to the reviewer’s concerns, by the order they appear.
>
> * It is true that a non-convex problem in general has many spurious local minima, and it is also true that in practice, researchers sometimes don’t observe such bad local minima. Understanding this seemingly contradictory phenomenon in its most general form is one of the fundamental open problems in deep learning.
> Recent results indicate that some important nonconvex problems satisfy this property, i.e., that all local minima are global. Examples include matrix completion, matrix sensing, PCA, etc. (cf. [1] and references therein). After the author in [2] showed that linear neural networks with squared error loss are examples of such problems, there have been efforts to extend this to more general neural networks. Our paper answers the question of "how far can we extend the *local min is global* property," by providing a general result (Theorem 4 & Corollary 5) on linear networks for which this property holds. More importantly, we also show that this property is not robust and immediately breaks when we add nonlinearities to the network. (Theorem 1 & 2).
>
> * Our focus in this paper was about the properties of the loss surface of empirical risk, disjoint from algorithms or architectural choices. Of course, in practice, current training methods are able to find better solutions than convex models, even with existence of many spurious local minima. The results of our paper suggest that (as mentioned briefly in Section 5), the loss surface itself is not likely to provide sufficient explanation of why optimization works well in deep learning, and studying the effect of algorithms and other factors will be a valuable direction of future work.
>
> [1] Ge et al., No Spurious Local Minima in Nonconvex Low Rank Problems: A Unified Geometric Analysis, 2017
> [2] Kawaguchi, Deep learning without poor local minima, 2016

---

### Official Review · AnonReviewer2 · 2018-11-07

**Rating:** 7
**Confidence:** 3

**Review:**

The authors present some theoretical results on the loss surface of neural networks. Their main results are:

(1) They consider a 1 layer hidden neural network where the single nonlinearity is ReLU / ReLU-like. Here they prove that as long as a linear model cannot fit the data, then there exits a local minimum strictly inferior to the global one (They can then scale the parameters to get infinitely many local optima).

The key idea is to construct a local minima whose risk value is the same as the local least squares solution. Then to construct a set of parameters that has smaller risk value than this local optima. The proof technique is interesting.

(2) They construct a particular dataset for which a one hidden layer neural net with other nonlinear activations (sigmoid, tanh, etc.) also has local optima.

I think this theorem is a bit less interesting since the dataset given has only two data points. I think it is less interesting to prove suboptimality of neural nets in small sample size settings.

(3) Global optimailty of linear networks. The authors show that deep linear networks (i.e. y = W1 W2 W3...W5 x) have only global minima or saddle points.

I'm not familiar enough with the field to know the significant of this result. The deep linear network  just seems like an artificial construction (i.e. in practice one would simply condense W1...W5 to one W) to study nonconvexity / local optima, no one would use it in practice.

---

> ### Author Response · Authors · 2018-11-22
> **Reply to AnonReviewer2**
>
> Thank you for your efforts in reviewing our paper. Below, we address the concerns raised, listed by the item number:
>
> (2) First, we point out a minor point: The dataset contains three data points. The input is two-dimensional, the output is one-dimensional, and there are three data points. We agree that suboptimality of neural nets with large sample size would be more interesting. While we are presenting a toy example to make a point, we would like to emphasize that this counterexample works for a *variety* of popular activation functions that are used in practice.
>
> (3) You are correct; a deep linear network is a purely theoretical subject of research, and it is not a practical model. However, the reason for studying it is that this simplified model can deliver some insights in understanding (nonlinear) neural networks. This model received a lot of attention since [Kawaguchi, NIPS’16] showed that linear neural networks with squared error loss have only global minima and saddle points, which implies that all local minima are globally optimal. There have been many efforts to extend this result to more general settings [e.g., Laurent & Brecht, ICML’18a,b]. Our paper settles the question of how far this property can be extended, and whether this can explain the fact that despite nonconvexity, gradient-based techniques can find weights that achieve near-zero loss. By providing a general result (Theorem 4 & Corollary 5) on linear neural networks for which this property holds, and also showing that this property fails to hold as soon as we add small nonlinearities to the hidden nodes (Theorem 1 & 2), we contribute to the growing body of literature that is trying to understand deep learning.

---

### Public Comment · (anonymous) · 2018-12-02
**Important Related Work**

For the realizable case, Liang et al. [1] have constructed spurious local minima for ReLU-class functions (sigma(x) = 0 for x <= 0, so include ELU), leaky-ReLU class functions (sigma(x)=x, for x>=0), quadratic-like functions and sigmoid-like functions. Each of them represents a class of activations that satisfying some conditions, with a similar flavor to (C2.1)-(C2.7)). Though the loss function is different (Liang et al. is for binary classification), this paper is not the first one to construct spurious local minima for many neuron activations. Could the authors comment on the differences?

[1] Liang, S., Sun, R., Li, Y., & Srikant, R. (2018). Understanding the loss surface of neural networks for binary classification. arXiv preprint arXiv:1803.00909.

---

> ### Author Response · Authors · 2018-12-04
> **Reply to the comment**
>
> Thank you for your interest in our paper, and also for pointing out a relevant result! The paper you noted is indeed related, but its focus is a bit different from our paper. The authors made assumptions on the loss function, data distribution, network structure, and activation function, and showed that all local minima of the empirical loss have zero classification error. Then, they presented counterexamples that if any of their assumptions are violated, there can be counterexamples such that local minima can have *non-zero classification error*. Please note that having nonzero classification error does not necessarily imply that the local minima are spurious, and having spurious local minima does not necessarily imply nonzero classification error either. Thus, the counterexamples do not directly relate to our work and the results are not directly comparable (as noted by the comment, the loss functions are also different). Having said that, we do appreciate the reference and we will add a citation in our next revision.
>
> In comparison, our paper shows that even for a single dataset, not taylor-made for specific activation functions, there can be a spurious local minimum for a variety of activation functions. In fact, a draft version of our paper appeared earlier than the paper mentioned by the comment, which is one of the reasons why we missed it in our paper. We promise to cite this paper and explain the differences in detail, in the next version of our paper.

---

> > ### Public Comment · (anonymous) · 2019-03-17
> > **Liang et al. does construct spurious local minima**
> >
> > A clarification: Liang et al. not only claimed that "there can be counterexamples such that local minima can have *non-zero classification error*", but also claimed that "these local minima are spurious" with high objective value (not just high classification error). In this sense, the nature of that paper is the same as your paper.
> >
> > Thanks for pointing out that your paper constructed one dataset that works for many neurons. This is different from the mentioned paper.
> >
> > BTW: Liang et al. also pointed out that quadratic loss may not be a good metric for binary classification since there exists global minimum with high classification error. This claim is irrelevant for pure optimization purpose but important for machine learning purpose.

---

### Meta-Review · Area_Chair1 · 2018-12-17
**Interesting investigation into the existence of spurious local minima in nonlinear networks**

**Confidence:** 5
**Recommendation:** Accept (Poster)

**Metareview:**

This is an interesting paper that develops new techniques for analyzing the loss surface of deep networks, allowing the existence of spurious local minima to be established under fairly general conditions.  The reviewers responded with uniformly positive opinions.